# Moner: Motion Correction in Undersampled Radial MRI with Unsupervised Neural Representation

**Qing Wu**[§]
ShanghaiTech University
wuqing@shanghaitech.edu.cn

**Chenhe Du**[§]
ShanghaiTech University
duchenhe@shanghaitech.edu.cn

**Xuanyu Tian**
ShanghaiTech University
tianxy@shanghaitech.edu.cn

**Jingyi Yu**
ShanghaiTech University
yujingyi@shanghaitech.edu.cn

**Yuyao Zhang**
ShanghaiTech University
zhangyy8@shanghaitech.edu.cn

**Hongjiang Wei**[¶]
Shanghai Jiao Tong University
hongjiang.wei@sjtu.edu.cn

## Abstract

Motion correction (MoCo) in radial MRI is a particularly challenging problem due to the unpredictability of subject movement. Current state-of-the-art (SOTA) MoCo algorithms often rely on extensive high-quality MR images to pre-train neural networks, which constrains the solution space and leads to outstanding image reconstruction results. However, the need for large-scale datasets significantly increases costs and limits model generalization. In this work, we propose **Moner**, an unsupervised MoCo method that jointly reconstructs artifact-free MR images and estimates accurate motion from undersampled, rigid motion-corrupted $k$-space data, without requiring any training data. Our core idea is to leverage the continuous prior of implicit neural representation (INR) to constrain this ill-posed inverse problem, facilitating optimal solutions. Specifically, we integrate a quasi-static motion model into the INR, granting its ability to correct subject's motion. To stabilize model optimization, we reformulate radial MRI reconstruction as a back-projection problem using the Fourier-slice theorem. Additionally, we propose a novel coarse-to-fine hash encoding strategy, significantly enhancing MoCo accuracy. Experiments on multiple MRI datasets show our Moner achieves performance comparable to SOTA MoCo techniques on in-domain data, while demonstrating significant improvements on out-of-domain data. The code is available at: **https://github.com/iwuqing/Moner**

## 1 Introduction

Radial magnetic resonance imaging (MRI) is an important technique in medical diagnostics and research (Feng, 2022), where measurements, *i.e.*, $k$-space data, are acquired by lines passing through the center of Fourier space from different views. However, its long acquisition time increases costs and makes it susceptible to motion artifacts (Spieker et al., 2023a). While undersampling can effectively accelerate MRI acquisition, reconstructing artifact-free MR images from undersampled, motion-corrupted $k$-space data is a challenging ill-posed inverse problem due to violation of Nyquist's sampling theorem and motion effects. Conventional analytical methods, such as NuIFFT (Fessler, 2007), lack effective prior constraints and often fail to recover satisfactory MR images, emphasizing the need for advanced motion correction (MoCo) algorithms.

---

[§]Equal contribution.
[¶]Corresponding author.

As the emergence of deep learning (DL), supervised methods have significantly improved the quality of MR images (Han et al., 2018; Liu et al., 2020; Sommer et al., 2020). They typically train deep neural networks on large-scale paired MRI datasets to learn inverse mappings from artifact-corrupted images to artifact-free ones. However, such end-to-end learning paradigm neglects motion modeling, causing severe hallucinations (Singh et al., 2024), where visually plausible MRI reconstructions are inconsistent with the acquired $k$-space data.

Recent works (Singh et al., 2024; Levac et al., 2023; 2024) propose integrating pre-trained neural networks with model-based optimization. Data priors from the pre-trained networks effectively constrain the solution space, while model-based optimization ensures reliable data consistency using MRI physical models, enabling the recovery of high-quality MR images with high data fidelity. However, pre-training the networks necessitates numerous diagnosis-quality MR images, significantly increasing reconstruction costs. Although these models demonstrate greater robustness compared to supervised methods, they remain vulnerable to out-of-domain (OOD) issues. These limitations undermine their practicality and reliability in real-world applications.

As an unsupervised DL framework, implicit neural representation (INR) has shown great promise in MRI reconstruction (Shen et al., 2022; Xu et al., 2023; Spieker et al., 2023b). The INR-based methods represent the unknown MR image as a continuous function parameterized by a multilayer perceptron (MLP). By incorporating the differential Fourier transform, the MLP network can be optimized by minimizing prediction errors on the $k$-space data. The learning bias of the MLP to low-frequency signals (Rahaman et al., 2019; Xu et al., 2019) facilitates recovery of high-quality MR images under non-ideal conditions. However, the potential of the unsupervised INR framework for the challenging MRI MoCo problem remains unexplored.

In this work, we propose **Moner**, a novel INR-based approach for addressing rigid motion artifacts in undersampled radial MRI. By leveraging the continuous prior inherent in the INR, our Moner can effectively tackle the ill-posed nature of the inverse problem, while presenting several innovative designs to enhance reconstructions. First, we introduce a quasi-static motion model into the INR, granting its ability to accurately estimate subject's rigid motion during MRI acquisition. Then, we reformulate the radial MRI recovery as a back-projection problem using the Fourier-slice theorem. This new formulation allows us to optimize the INR model on projection data (*i.e.*, Radon transform), mitigating the high dynamic range issues caused by the MRI $k$-space data and thus stabilizing model optimization. Moreover, based on hash encoding technique (Müller et al., 2022), we introduce a novel coarse-to-fine learning strategy, significantly improving MoCo accuracy.

The proposed Moner is an unsupervised DL model, making it adaptable to various MRI scenarios. We evaluate its performance on two public MRI datasets, including fastMRI (Knoll et al., 2020) and MoDL (Aggarwal et al., 2018). The results show our method performs comparably to state-of-the-art MoCo techniques on in-domain data, while significantly surpassing them on out-of-domain data. Extensive ablation studies validate the effectiveness the several key components of our Moner.

## 2 PRELIMINARIES AND RELATED WORKS

In this section, we first revisit the Fourier-slice theorem to discuss the relationship between radial $k$-space data and projection data (Sec. §2.1). Then, we briefly review advanced approaches for MRI motion correction (Sec. §2.2) and implicit neural representation (Sec. §2.3).

### 2.1 RADIAL MRI $k$-SPACE DATA VERSUS PROJECTION DATA

Let $\boldsymbol{f}(x,y) \in \mathbb{C}^{h \times w}$ represent a complex-valued MR image of $h \times w$ size, then the radial MRI $k$-space data $\boldsymbol{k}(\theta,\omega) \in \mathbb{C}^{n \times m}$ can be written as:

$$\boldsymbol{k}(\theta,\omega) = \iint \boldsymbol{f}(x,y) \cdot \mathrm{e}^{-j2\pi\omega(x\cos\theta + y\sin\theta)} \mathrm{d}x\mathrm{d}y, \tag{1}$$

where $\theta \in [0, 2\pi)$ represents the acquisition angle. $n$ and $m$ respectively denote the number and size of lines (*i.e.*, $\boldsymbol{k}(\theta, \cdot) \in \mathbb{C}^m$, also known as spokes) in the $k$-space data.

According to the Fourier-slice theorem (Gonzalez, 2009), the 1D inverse Fourier transform (IFT) of the $k$-space data $\boldsymbol{k}(\theta,\omega)$ over the variable $\omega$ is equal to an integral projection $\boldsymbol{g}(\theta,\rho) \in \mathbb{C}^{n \times m}$ of the

MR image along the view $\theta$, defined as:

$$
\begin{aligned}
\boldsymbol{g}(\theta, \rho) &\triangleq \iint \boldsymbol{f}(x, y) \cdot \delta(x \cos \theta + y \sin \theta - \rho) \mathrm{d}x \mathrm{d}y \\
&= \int \underbrace{\iint \boldsymbol{f}(x, y) \cdot \mathrm{e}^{-j2\pi\omega(x \cos \theta + y \sin \theta)} \mathrm{d}x \mathrm{d}y}_{\boldsymbol{k}(\theta, \omega)} \cdot \mathrm{e}^{j2\pi\omega\rho} \mathrm{d}\omega = \mathcal{T}_\omega^{-1}\{\boldsymbol{k}(\theta, \omega)\},
\end{aligned}
\tag{2}
$$

where $\mathcal{T}_\omega^{-1}\{\cdot\}$ denotes the 1D IFT operator over the variable $\omega$ and $\delta(\cdot)$ is the Dirac delta function. Eq. 2 implies that the radial MRI solving (*i.e.*, $\boldsymbol{k} \to \boldsymbol{f}$) can be theoretically reformulated as a back-projection problem (*i.e.*, $\boldsymbol{g} \to \boldsymbol{f}$) using the 1D IFT operator. A recent study on cardiac MRI acceleration (Catalán et al., 2023) introduces the Fourier-slice theorem to optimize a neural field directly from raw *k*-space data, showing promising potential. Similarly, but with a distinct focus, our work uses the Fourier-slice theorem to reformulate radial MRI as a back-projection problem, fundamentally addressing the high-dynamic range issue and stabilizing optimization.

## 2.2 ADVANCED APPROACHES FOR MRI MOTION CORRECTION

Classical optimization methods use explicit priors, such as TV for local smoothness (Rudin et al., 1992), to address the MRI MoCo problem. However, these handcrafted regularizers fail to fully characterize distribution of images, leading limited performance. With the advancement of high-performance computing and neural networks, DL-based methods (Han et al., 2018; Küstner et al., 2019; Liu et al., 2020; Sommer et al., 2020; Duffy et al., 2021; Lyu et al., 2021; Bao et al., 2022; Lee et al., 2021; Oksuz, 2021; Haskell et al., 2019; Cui et al., 2023; Chen et al., 2023a; Singh et al., 2024; Levac et al., 2023; 2024; Klug et al., 2024) have significantly outperformed traditional model-based optimization algorithms. Currently, the Score-MoCo proposed by Levac et al. (2023; 2024) is the state-of-the-art (SOTA) MRI MoCo method, benefiting from generative diffusion priors and well-designed optimization strategies. However, these DL-based methods often require extensive high-quality MR images for pre-training neural networks, which significantly increases reconstruction costs and often suffers from the out-of-domain (OOD) problem (Spieker et al., 2023a). In contrast, our Moner follows an unsupervised paradigm and does not require additional MRI data, significantly enhancing its applicability and generalization across various clinical scenarios.

## 2.3 IMPLICIT NEURAL REPRESENTATION FOR MRI RECONSTRUCTION

Implicit neural representation (INR) has emerged as a universal unsupervised DL framework for visual inverse problems, such as novel view synthesis (Mildenhall et al., 2021) and surface reconstruction (Wang et al., 2021). Its core concept is to leverage the learning bias inherent in neural networks (Rahaman et al., 2019; Xu et al., 2019) to regularize the underdetermined nature of inverse problems, while incorporating differentiable forward models to simulate physical processes for high data fidelity. Recently, many INR-based MRI reconstruction methods (Feng et al., 2023; Shen et al., 2022; Xu et al., 2023; Kunz et al., 2023; Feng et al., 2022; Spieker et al., 2023b; Wu et al., 2021; Huang et al., 2023; Chen et al., 2023c; Catalán et al., 2023) have been proposed. By using Fourier transforms to simulate MRI acquisition, they can reconstruct high-quality MR images with high data fidelity. However, these INR-based methods fail to address motion-corrupted radial MRI, since they lack motion modeling. Moreover, these methods either entirely overlook the high dynamic range problem in *k*-space data or specifically design a relative $\ell_2$ loss to alleviate it (Huang et al., 2023; Spieker et al., 2023b; Feng et al., 2022). Instead, our Moner incorporates a motion model into the INR framework and introduces a new formulation for the radial MRI, fundamentally addressing the MRI MoCo problem.

## 3 PROPOSED METHOD

This section introduces our Moner model. First, we define a new formulation for the rigid motion-corrupted radial MRI (Sec. §3.1). Then, we present a quasi-static motion model within the INR framework and our model optimization pipeline (Sec. §3.2). Finally, we propose a novel coarse-to-fine hash encoding that can significantly improve MoCo accuracy (Sec. §3.3). An overview of the proposed Moner model is shown in Fig. 1.

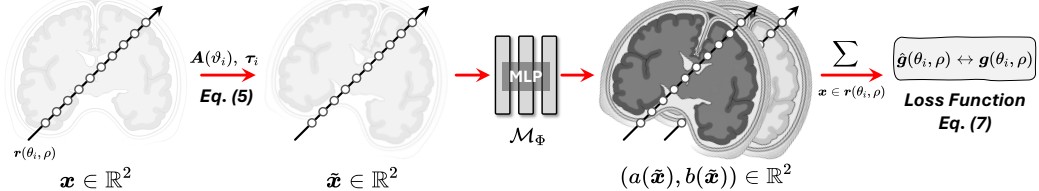

Figure 1: Overview of proposed Moner model. Given any ray $r(\theta_i, \rho)$ at the 2D canonical space, we uniformly sample multiple coordinates $x \in r(\theta_i, \rho)$ and generate their version $\tilde{x}$ at the physical-world space via spatial transform (Eq. 5). Then, the INR network $\mathcal{M}_\Phi$ predicts the real $a(\tilde{x})$ and imaginary $b(\tilde{x})$ parts of the MR images. The projection data $\hat{g}(\theta_i, \rho)$ can be obtained using integral projection (Eq. 6). Finally, we jointly optimize the INR $\mathcal{M}_\Phi$ and motion parameters $\{\vartheta_i, \tau_i\}$ minimizing the loss $\mathcal{L}$ (Eq. 7) between the estimated $\hat{g}(\theta_i, \rho)$ and measured $g(\theta_i, \rho)$ projections.

## 3.1 PROBLEM FORMULATION

Our goal is to reconstruct artifact-free MR images $f(x, y)$ from undersampled (*i.e.*, $nm < hw$ in Eq. 1), rigid motion-corrupted radial MRI $k$-space data $k(\theta, \rho)$ in *an unsupervised manner*. There are two key challenges in solving the ill-posed inverse problem: 1) How to inject effective priors for narrowing the solution space? 2) How to ensure stable optimization solving?

The SOTA Score-MoCo method (Levac et al., 2023; 2024) represents the MR image as a discrete matrix and uses diffusion priors to solve it, requiring numerous MRI images for pre-training and suffer from the OOD problem. Our Moner instead adopts the inherent prior of the unsupervised INR to learn the continuous representation of the MR image, in which an implicit function of spatial coordinates for the complex-valued MR image is defined as below:

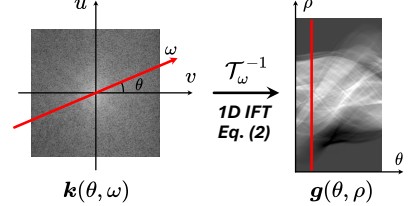

Figure 2: Illustration of transforming radial MRI $k$-space data $k$ into projection data $g$ via the 1D IFT $\mathcal{T}_\omega^{-1}$.

$$f : x = (x, y) \in \mathbb{R}^2 \longrightarrow (a(x), b(x)) \in \mathbb{R}^2, \quad (3)$$

where $x$ denotes any spatial coordinate in a 2D canonical space $[-1, 1] \times [-1, 1]$. The variables $a(x)$ and $b(x)$ represent the real and imaginary parts of the complex-valued MRI image $f(x)$ at position $x$. We then use an MLP network $\mathcal{M}_\Phi$, which takes any coordinate $x$ as input and outputs a 2D vector corresponding to the real and imaginary parts $(a(x), b(x))$, to fit the function $f$. Due to the neural networks' inherent learning bias towards low-frequency signal patterns (Rahaman et al., 2019; Xu et al., 2019), this continuous function $f$ can be well-approximated, enabling the recovery of high-quality MR images.

To achieve stable inverse learning, we introduce the Fourier-slice theorem (Eq. 2) to reformulate radial MRI reconstruction (*i.e.*, $k \to f$) as a back-projection problem (*i.e.*, $g \to f$). As illustrated in Fig. 2, we first transform the $k$-space data $k$ into projection data $g$ using the 1D IFT operator $\mathcal{T}_\omega^{-1}$. By employing a differentiable projection model (Eq. 6), we then optimize the MLP network $\mathcal{M}_\Phi$ to reconstruct MR images by minimizing prediction errors on the projection data $g$. Since the value range of the projection data $g$ is much narrower than that of the $k$-space data $k$, this optimization solving effectively avoids the high dynamic range problem (Feng et al., 2022; Spieker et al., 2023b) and thus stabilizes model optimization.

## 3.2 QUASI-STATIC MOTION MODEL AND OPTIMIZATION

We introduce a quasi-static motion model (Spieker et al., 2023a), a standard model in the field of the MoCo, into the INR framework, extending it for estimating the subject's motion. The motion model has two basic assumptions: 1) The motion is rigid, which is common in many clinical MRI acquisitions, such as brain and leg imaging. 2) The motion occurs between the spokes but remains stationary during the fast acquisition of a single spoke. This quasi-static assumption is reasonable since scanning a single spoke typically only takes a few milliseconds using radial MRI sequences, such as fast low-angle shot (Zhang et al., 2010).

For any acquisition views $\theta_i$, where $i \in \{1, 2, \cdots, n\}$, we define a corresponding learnable motion triplet $(\vartheta_i, \tau_{x,i}, \tau_{y,i})$, where $\vartheta_i$ denotes rotation angle, and $\tau_{x,i}$ and $\tau_{y,i}$ are shifts along the X-axis and Y-axis. Then, a rotation matrix $\boldsymbol{A}(\vartheta_i) \in \mathbb{R}^{2\times2}$ and shift vector $\boldsymbol{\tau}_i \in \mathbb{R}^2$ are defined as:

$$\boldsymbol{A}(\vartheta_i) = \begin{pmatrix} \cos\vartheta_i & -\sin\vartheta_i \\ \sin\vartheta_i & \cos\vartheta_i \end{pmatrix}, \quad \boldsymbol{\tau}_i = \begin{pmatrix} \tau_{x,i} \\ \tau_{y,i} \end{pmatrix}, \tag{4}$$

where the motion triplets $\{(\vartheta_i, \tau_{x,i}, \tau_{y,i})\}_{i=1}^n$ are estimated from scratch.

Fig. 1 demonstrates the workflow to optimize our Moner model. Given the projection data $\boldsymbol{g}(\theta_i, \rho)$, we first construct a ray $\boldsymbol{r}(\theta_i, \rho) = \{(x, y) \mid x\cos\theta_i + y\sin\theta_i = \rho\}$ in the 2D canonical space and uniformly sample a set of coordinates $\boldsymbol{x} \in \boldsymbol{r}(\theta_i, \rho)$ by a pre-defined interval $\Delta\boldsymbol{x}$. Then, these coordinates are transformed into a physical-world space (*i.e.*, with the presence of the rigid motion) via a spatial transformation operator, defined as:

$$\tilde{\boldsymbol{x}} = \boldsymbol{A}(\vartheta_i)\boldsymbol{x} + \boldsymbol{\tau}_i, \quad \forall \boldsymbol{x} \in \boldsymbol{r}(\theta_i, \rho), \tag{5}$$

where $\tilde{\boldsymbol{x}}$ represent the coordinates at the physical-world space. Then, the MLP network $\mathcal{M}_\Phi$ takes these coordinates $\tilde{\boldsymbol{x}}$ as inputs and predicts the corresponding real and imaginary parts $(a(\tilde{\boldsymbol{x}}), b(\tilde{\boldsymbol{x}})) = \mathcal{M}_\Phi(\tilde{\boldsymbol{x}})$ of the MR image in the real-world space, *i.e.*, with the subject's motion. Finally, we can generate the projection data using a integral projection model, defined by

$$\hat{\boldsymbol{g}}(\theta_i, \rho) = \sum_{\boldsymbol{x}\in\boldsymbol{r}(\theta_i,\rho)} a(\tilde{\boldsymbol{x}}) \cdot \Delta\boldsymbol{x} + j \sum_{\boldsymbol{x}\in\boldsymbol{r}(\theta_i,\rho)} b(\tilde{\boldsymbol{x}}) \cdot \Delta\boldsymbol{x}. \tag{6}$$

Because both the spatial transformation (Eq. 5) and integral projection model (Eq. 6) are differentiable, we can jointly optimize the motion triplets $\{(\vartheta_i, \tau_{x,i}, \tau_{y,i})\}_{i=1}^n$ and the MLP network $\mathcal{M}_\Phi$ using back-propagation techniques, such as Adam (Kingma & Ba, 2014), to minimize the loss function $\mathcal{L}$ measuring errors between the estimated and real projections, defined as:

$$\mathcal{L} = \sum_{\boldsymbol{r}(\theta_i,\rho)\in\mathcal{R}} \left( \left| \Re\{\hat{\boldsymbol{g}}(\theta_i, \rho) - \boldsymbol{g}(\theta_i, \rho)\} \right| + \left| \Im\{\hat{\boldsymbol{g}}(\theta_i, \rho) - \boldsymbol{g}(\theta_i, \rho)\} \right| \right), \tag{7}$$

where $\mathcal{R}$ is a set of the random sampling ray $\boldsymbol{r}(\theta_i, \rho)$ at each optimization step. $\Re\{\cdot\}$ and $\Im\{\cdot\}$ respectively are the real and imaginary parts of a complex value. After the model optimization, the high-quality MR image $\boldsymbol{f} = a + jb$ can be solved by feeding all voxel coordinates $\boldsymbol{x}$ at the 2D canonical space into the well-trained network $\mathcal{M}_\Phi$.

### 3.3 Coarse-to-fine Hash Encoding

We use cutting-edge hash encoding (Müller et al., 2022) with two fully-connected layers to implement the MLP network $\mathcal{M}_\Phi$. The hash encoding module transforms low-dimensional coordinates $\boldsymbol{x}$ into high-dimensional features $\{\mathbf{v}_i(\boldsymbol{x}) \in \mathbb{R}^F\}_{i=1}^L$ at multiscale resolutions of $L$. This mapping significantly enhances the MLP network's ability to fit high-frequency signals, thereby accelerating model optimization. However, the powerful hash encoding always impedes motion estimation, ultimately degrading model performance. This is because solving motion estimation primarily relies on low-frequency structural patterns (*e.g.*, skull and leg bones) rather than high-frequency details (*e.g.*, cerebellum). While using a coarse-resolution hash encoding with a small $L$ can improve motion estimation by focusing on these low-frequency signals, it also limits the model's capacity to capture high-frequency components, leading to a loss of image details.

To this end, we propose a novel coarse-to-fine learning strategy to achieve the balance between motion estimation and fine-detailed image reconstruction. Technically, we first assign $L$ mask vectors $\{\boldsymbol{\alpha}_i\}_{i=1}^L$ for the features $\{\mathbf{v}_i(\boldsymbol{x})\}_{i=1}^L$ at the multiscale resolutions. Inspired by the studies of camera registrations (Lin et al., 2021; Chen et al., 2023b), we piecewise update the mask vectors during the model optimization as follow:

$$\boldsymbol{\alpha}_i = \begin{cases} \mathbf{1}^\top \in \mathbb{R}^F & \text{if } i < \lambda \\ \mathbf{0}^\top \in \mathbb{R}^F & \text{else} \end{cases}, \quad i = 1, 2, \cdots, L, \tag{8}$$

where $\lambda \in [0, L]$ is a value proportional to the optimization process. Leveraging the mask vectors $\{\boldsymbol{\alpha}_i\}_{i=1}^L$, we can control periodically the spatial resolution of the hash encoding. Finally, the output feature $\mathbf{v} \in \mathbb{R}^{LF}$ is generated by

$$\mathbf{v}(\boldsymbol{x}) = \{\mathbf{v}_1(\boldsymbol{x}) \odot \boldsymbol{\alpha}_1\} \oplus \{\mathbf{v}_2(\boldsymbol{x}) \odot \boldsymbol{\alpha}_2\} \cdots \oplus \{\mathbf{v}_L(\boldsymbol{x}) \odot \boldsymbol{\alpha}_L\}, \tag{9}$$

where $\odot$ denotes the Hadamard product and $\oplus$ denotes the concatenation operator.

Through the coarse-to-fine learning strategy, in early optimization our model can only capture the low-frequency global structures due to the coarse resolution hash encoding, which benefits accurate motion estimation. As the iteration continues, its learning ability gradually improves, enabling it to recover high-frequency local image details.

## 4 EXPERIMENTS

This section explores two key questions: 1) Can our unsupervised Moner outperform the SOTA techniques for the rigid motion-corrupted MRI reconstruction? 2) How do the key components of our Moner affect its performance? We conduct extensive experiments to address these questions.

### 4.1 EXPERIMENTAL SETUPS

**Datasets**  The fastMRI (Knoll et al., 2020) and MoDL (Aggarwal et al., 2018) datasets are used in our experiments. For the fastMRI dataset, we first extract 1,925 brain MR slices with image sizes of 320×320 and an image spacing of 1×1 mm$^2$ from different subjects along the axial direction. These slices include three contrasts of T1w, T2w, and FLAIR. We then split them into three parts, including 1,800 slices for training set, 100 for validation set, and 25 for test set. The training and validation sets are used solely for training supervised baselines, while our Moner does not access them. For the MoDL dataset, we use 20 T2w brain MR slices with image sizes of 256×256 and an image spacing of 1×1 mm$^2$ from 20 subjects along the sagittal direction for additional test.

**Pre-processing**  We use a 2D radial sampling pattern with the golden-angle acquisition scheme. Each spoke has a length of 511, corresponding to an imaging FOV of 511×511 mm$^2$. The fully sampled radial $k$-space data thus consists of approximately 720 views. For undersampled MRI, we set acceleration factors (AFs) to 2× and 4×, corresponding to 360 and 180 views of undersampled radial $k$-space data, respectively. Following the motion simulation pipeline used in previous studies (Levac et al., 2023; 2024; Spieker et al., 2023a), we first divide all spokes into 18 motion stages, where spokes within the same motion stage share the same motion trajectory. Then, we simulate four levels of rigid motion $\beta = \{2, 5, 10, 15\}$ with random rotations of $[-\beta, \beta]°$ and shifts of $[-\beta, \beta]$ mm along the X-axis and Y-axis. *The MRI reconstructions under different settings (i.e., different AFs and motion ranges $\beta$) are considered as different tasks. Thus, all training and test processes are conducted independently.*

**Baselines and Metrics**  Four representative MRI MoCo approaches are compared, including: 1) one analytical algorithm (NuIFFT (Fessler, 2007)), 2) one iterative optimization algorithm (TV (Rudin et al., 1992)), 3) one image-based supervised DL models (DRN-DCMB (Liu et al., 2020)), and 4) one supervised diffusion-based DL method (Score-MoCo (Levac et al., 2023; 2024)). The Score-MoCo is currently the SOTA model for the MRI MoCo. For the reconstructed MR images, we use peak signal-to-noise ratio (PSNR) and structural similarity index (SSIM) as quantitative evaluation metrics. For the estimated motion parameters, we compute the standard deviation of absolute errors between estimations and trues, denoted by $\sigma_\vartheta$ and $\sigma_\tau$. *More details of the baselines and metrics can be found in Appendix A.2 and A.3.*

**Implementation Details**  For our Moner, we employ the hash encoding (Müller et al., 2022) followed by two fully-connected (FC) layers with a width of 128 to implement the MLP network $\mathcal{M}_\Phi$. The first FC layer is followed by a ReLU activation, while the second one (i.e., , the output layer) has no activation. For the hash encoding (Müller et al., 2022) used in our model, we set its hyper-parameters as follows: base resolution $N_{min} = 2$, maximal hash table size $T = 2^{18}$, and resolution growth rate $b = 2$. Our coarse-to-fine strategy sets its resolution $L$ from 4 to 16 as optimization progresses. At each iteration, we randomly sample 80 rays (i.e., $|\mathcal{R}| = 80$ in Eq. 7). We use the Adam algorithm with default hyper-parameters (Kingma & Ba, 2014) to optimize the model. The learning rate is initialized to 0.001 and decays by half every 1,000 epochs. The total number of epochs is 4,000. *Note that here the hyper-parameters are determined based on 5 samples from the training set of the fastMRI dataset (Knoll et al., 2020) and are kept consistent across all other cases.*

Table 1: Quantitative results (Mean in $\sigma_\vartheta/\sigma_\tau$) of motion parameters by compared methods on the fastMRI and MoDL datasets. Results of t-test statistical tests comparing our Moner to baselines are denoted by ** ($p$-value $< 0.01$), * ($p$-value $< 0.05$), and ▼ (not significant, $p$-value $\geq 0.05$). Here the "AF" and "MR" represent acceleration rate and motion range, respectively. The best performances are highlighted in **bold**.

| Dataset | AF | MR | Optim. TV | Self-sup. Score-MoCo | Unsup. Moner (Ours) |
|---|---|---|---|---|---|
| fastMRI | 2× | ±2 | 0.038**/0.290** | 0.038**/0.293** | **0.009/0.057** |
| | | ±5 | 0.038**/0.309** | 0.038**/0.313** | **0.010/0.131** |
| | | ±10 | 0.037**/0.407* | 0.040**/0.470* | **0.009/0.305** |
| | | ±15 | 0.036**/0.621▼ | 0.044**/0.648▼ | **0.010/0.602** |
| | 4× | ±2 | 1.514**/0.676** | 0.052▼/0.312** | **0.034/0.060** |
| | | ±5 | 1.369**/0.670** | 0.062▼/0.296** | **0.047/0.119** |
| | | ±10 | 1.699**/0.922** | 0.047▼/0.411** | **0.042/0.279** |
| | | ±15 | 2.513**/1.520** | 0.050▼/0.532* | **0.043/0.812** |
| MoDL | 2× | ±2 | 0.037**/0.275** | 0.389▼/0.409** | **0.009/0.058** |
| | | ±5 | 0.040**/0.276** | 0.838*/0.720** | **0.009/0.144** |
| | | ±10 | 0.042**/0.371▼ | 0.748*/1.263** | **0.008/0.286** |
| | | ±15 | 0.040**/0.553▼ | 0.096**/1.448** | **0.009/0.570** |
| | 4× | ±2 | 0.167▼/0.348** | 1.946**/0.824** | **0.019/0.065** |
| | | ±5 | 0.276*/0.377** | 1.125**/0.782** | **0.021/0.163** |
| | | ±10 | 0.257▼/0.494* | 1.642*/1.242** | **0.022/0.324** |
| | | ±15 | 0.506*/0.824▼ | 1.721*/1.797** | **0.019/0.578** |

## 4.2 MAIN RESULTS

**Comparison with SOTAs for MoCo Accuracy** Table 1 compares the performance of our Moner with two baselines. The other two baselines are excluded as they do not explicitly model motion. The TV algorithm, relying on local image smoothness, performs well at a high sampling rate of AF = 2×. However, when the AF is increased to 4×, further worsening the ill-posed nature of the inverse problem, its performance significantly declines. The Score-MoCo model, pre-trained on the fastMRI dataset, delivers good MoCo results on the same datasets, but struggles with the OOD problem on the unseen MoDL dataset, performing even worse than the TV algorithm. In contrast, our Moner achieves the highest MoCo accuracy on both datasets, benefiting from the robust continuous prior of the INR and well-designed optimization strategy. Moreover, our method maintains stable performance across different MRs. For instance, the shift error $\sigma_\tau$ remains below 1 for MRs ranging from ±2 to ±15 on the fastMRI dataset. In summary, the results above confirm the superiority of Moner over SOTA techniques in our MoCo accuracy.

**Comparison with SOTAs for MR Image** Table 2 presents the quantitative comparisons (PSNR) of MR images reconstructed by our Moner and the baselines. The SSIM results are provided in Table 7. On the fastMRI dataset, both Score-MoCo and our Moner produce comparable and satisfactory results, significantly outperforming the other baselines. However, when applied to the unseen MoDL dataset, the Score-MoCo model suffers from severe performance degradation due to the OOD problem, even falling behind the TV algorithm at AF = 4×. This performance trend in MR image reconstruction mirrors the results in MoCo accuracy. In contrast, our Moner consistently achieves robust and high-quality reconstructions across both datasets. Fig. 4 shows the qualitative results. The analytical NuIFFT produces severe artifacts. The image-based supervised DRN-DCMB yields smooth but fuzzy MRI results. The TV algorithm, relying on local image smoothness,

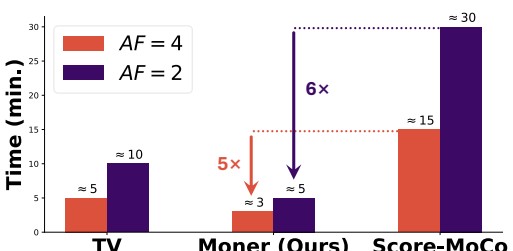

Figure 3: Speed comparison of TV, our Moner, and SOTA diffusion-based Score-MoCo on the fastMRI dataset.

Table 2: Quantitative results (Mean±STD in PSNR) of MR images by compared methods on the fastMRI and MoDL datasets. Results of t-test statistical tests comparing our Moner to baselines are denoted by ** ($p$-value $< 0.01$), * ($p$-value $< 0.05$), and ▼ (not significant, $p$-value $\geq 0.05$). Here the "AF" and "MR" represent acceleration rate and motion range, respectively. The best and second performances are highlighted in **bold** and underline, respectively.

| Dataset | AF | MR | Analy. | Optim. | Sup. | | Unsup. |
|---|---|---|---|---|---|---|---|
| | | | NuIFFT | TV | DRN-DCMB | Score-MoCo | Moner (Ours) |
| fastMRI | 2× | ±2 | 27.16±2.33** | 29.53±2.24** | 29.80±2.74** | **33.94±2.29**▼ | 32.64±2.65 |
| | | ±5 | 22.93±2.06** | 29.53±2.19** | 25.63±2.57** | **33.64±2.22**▼ | 32.62±2.61 |
| | | ±10 | 20.46±2.02** | 29.46±2.27** | 22.62±2.40** | **33.52±2.55**▼ | 32.49±2.78 |
| | | ±15 | 19.55±2.07** | 29.44±2.28** | 21.07±2.13** | **33.69±2.38**▼ | 32.50±2.65 |
| | 4× | ±2 | 25.99±2.01** | 25.12±3.91** | 29.18±2.44** | **32.33±2.39**▼ | 31.49±2.51 |
| | | ±5 | 22.30±2.15** | 25.56±3.72** | 25.18±2.53** | **32.03±2.36**▼ | 31.07±2.58 |
| | | ±10 | 19.81±1.96** | 25.41±4.05** | 22.01±2.36** | **32.34±2.25**▼ | 31.20±2.58 |
| | | ±15 | 19.20±2.03** | 25.12±3.98** | 21.22±2.23** | **32.22±2.38**▼ | 31.07±2.74 |
| MoDL | 2× | ±2 | 28.51±1.19** | 31.33±1.11** | 29.66±1.04** | 33.34±3.54▼ | **34.56±0.92** |
| | | ±5 | 25.15±0.96** | 31.32±1.11** | 25.79±0.73** | 32.45±4.27* | **34.54±0.90** |
| | | ±10 | 23.31±1.05** | 31.29±1.11** | 23.98±1.10** | 32.30±3.53* | **34.57±0.96** |
| | | ±15 | 22.89±0.97** | 31.26±1.07** | 22.52±1.04** | 33.90±1.28▼ | **34.32±0.91** |
| | 4× | ±2 | 27.59±1.08** | 30.70±1.51** | 28.78±1.12** | 28.45±3.04** | **33.50±1.03** |
| | | ±5 | 24.39±0.95** | 30.30±2.00** | 25.67±0.91** | 30.09±2.55** | **33.24±0.88** |
| | | ±10 | 22.91±1.01** | 30.26±1.93** | 23.98±0.87** | 30.33±3.48** | **33.21±1.06** |
| | | ±15 | 22.40±0.92** | 29.81±1.87** | 22.72±0.96** | 29.94±3.19** | **33.25±0.81** |

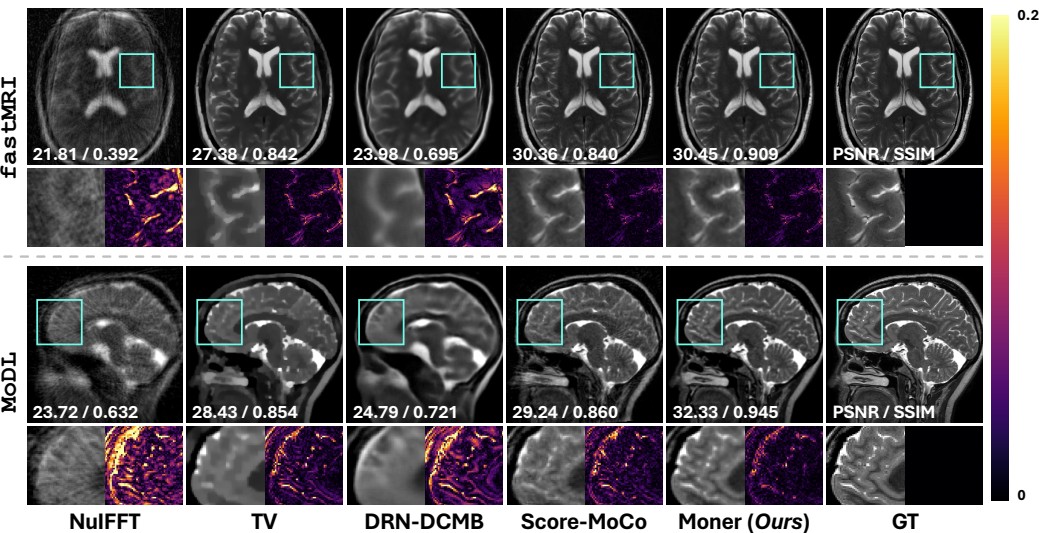

NuIFFT     TV     DRN-DCMB     Score-MoCo     Moner (*Ours*)     GT

Figure 4: Qualitative and quantitative results of MR images by compared methods on two test samples (#22 and #9) of the fastMRI and MoDL datasets for AF = 2× and MR = ±5.

reduces motion artifacts but introduces strong cartoon-like features and loses many image details. While Score-MoCo recovers high-quality MR images on the fastMRI dataset, its reconstructions on the unseen MoDL dataset contain many artifacts due to inaccurate MoCo cuased by the OOD problem. Visually, our Moner consistently achieves high-quality MRI reconstructions across both datasets, further confirming its robustness and superiority over SOTA MoCo methods. Fig. 3 compares the reconstruction speeds of our method with two iterative methods (TV and Score-MoCo) using a single NVIDIA RTX 4070 Ti GPU on the fastMRI dataset. Our Moner achieves the fastest reconstruction speed, being more than 5× and 6× faster than the SOTA Score-MoCo model. *Additional visual results are provided in Appendix A.4.*

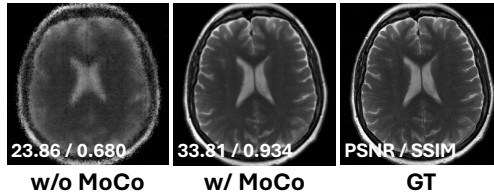

**w/o MoCo   w/ MoCo   GT**

Figure 5: Qualitative results of MR images by our Moner ablating the motion model on a sample (#5) the fastMRI dataset for AF = 2× and MR = ±5.

Table 3: Quantitative results of MR images by our Moner ablating the motion model on the fastMRI dataset for AF = 2× and MR = ±5. Results of t-test statistical tests are denoted by ** ($p$-value < 0.01), * ($p$-value < 0.05), and ▼ (not significant, $p$-value ≥ 0.05).

| Motion Model | PSNR | SSIM |
|---|---|---|
| w/o MoCo | 21.90±2.22** | 0.614±0.106** |
| w/ MoCo | **32.37±1.94** | **0.935±0.014** |

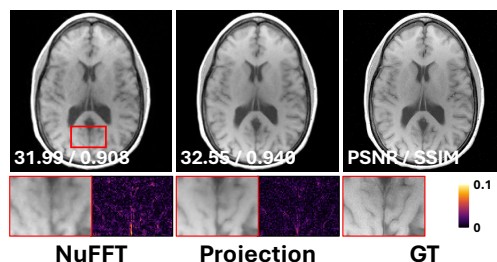

**NuFFT   Projection   GT**

Figure 6: Qualitative results of MR images by our Moner with different forward models on a sample (#4) the fastMRI dataset for AF = 2× and MR = ±5.

Table 4: Quantitative results of MR images and motion parameters by our Moner with different forward models on the fastMRI dataset for AF = 2× and MR = ±5. Results of t-test statistical tests are denoted by ** ($p$-value < 0.01), * ($p$-value < 0.05), and ▼ ($p$-value ≥ 0.05).

| Forw. Model | MR Image | Motion |
|---|---|---|
| | **PSNR** | $\sigma_\vartheta/\sigma_\tau$ |
| NuFFT | 31.38±2.11▼ | 0.108**/0.456** |
| Projection | **32.37±1.94** | **0.009/0.041** |

## 4.3 ABLATION STUDIES

**Influence of Quasi-static Motion Model**   We first explore the effectiveness of the quasi-static motion model in our Moner. Specifically, we remove the motion model while keeping other settings unchanged for a fair comparison. Fig. 5 shows the reconstructed MR images. Clearly, without the motion model, our Moner fails to produce satisfactory reconstructions, with many artifacts caused by motion. In contrast, the Moner with the motion model reconstructs clean images with fine details. Table 3 presents the quantitative results, showing that the motion model contributes to a significant improvement of over 10 dB in PSNR and 0.3 in SSIM. This ablation study demonstrates that the motion model plays an indispensable role in the motion-corrupted MRI problem.

**Influence of Forward Model for Optimization**   We then explore the influence of the forward model used in our optimization process. Current MRI MoCo methods (Singh et al., 2024; Levac et al., 2023; 2024) typically leverage NuFFT (Fessler, 2007) as the forward model for optimization, which leads to high dynamic range problems. Some additional optimization tricks (*e.g.*, normalizing the loss function) are required (Feng et al., 2022; Spieker et al., 2023b). In contrast, our Moner introduces the Fourier-slice theorem to reformulate MRI reconstruction as a back-projection problem, allowing the use of a differential projection model. As shown in Table 4, the projection model outperforms NuFFT in both MoCo accuracy and MRI reconstruction quality. The reconstructed MR images are shown in Fig. 6. From the visual comparison, the projection model produces superior image details compared to the NuFFT model. This ablation study confirms the effectiveness of the proposed projection-based optimization pipeline in improving model performance

**Influence of Coarse2fine Hash Encoding**   We finally investigate the influence of coarse-to-fine hash encoding on model performance. We compare it with the naive hash encoding at coarse ($L$=6) and fine ($L$=16) resolutions, keeping other model configurations the same for a fair evaluation. Fig. 8 shows the estimated motion parameters. Both the naive coarse ($L$=6) hash encoding and our coarse-to-fine strategy accurately predict motion parameters. However, the naive fine ($L$=16) hash encoding fails to correct the motion, confirming our argument that low-frequency image information is more crucial for solving MoCo. Fig. 7 displays the reconstructed MR images. The naive hash

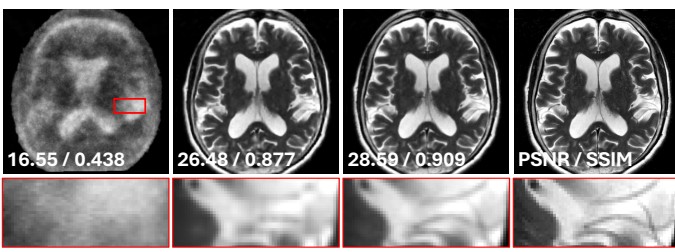

Figure 7: Qualitative results of MR images by our Moner with different hash encodings on a sample (#2) the fastMRI dataset for AF = 2× and MR = ±5.

Table 5: Quantitative results of MR images by our Moner with different hash encodings on the fastMRI dataset for AF = 2× and MR = ±5. Results of t-test statistical tests are denoted by ** ($p$-value < 0.01), * ($p$-value < 0.05), and ▼ ($p$-value ≥ 0.05).

| Hash Encoding | PSNR |
|---|---|
| Fine ($L$=16) | 23.69±3.82* |
| Coarse ($L$=16) | 30.77±2.21▼ |
| Coarse2fine | **32.37±1.94** |

encoding (both fine and coarse) cannot produce satisfactory MR images. The fine encoding degrades reconstruction due to incorrect motion estimation, while the coarse encoding loses image details due to its limited learning capacity. In contrast, our coarse-to-fine strategy achieves excellent visual results. The quantitative results shown in Table 5 also demonstrate the superiority of our method over the naive hash encoding.

# 5 CONCLUSION

This work proposes Moner, a novel method to address rigid motion-corrupted, undersampled radial MRI reconstruction. The proposed Moner is an unsupervised DL model, which eliminates the need for external training data and thus can flexibly adapt to different MRI acquisition protocols, such as different acceleration rates. Our Moner makes several key innovations that significantly improve MRI reconstructions, including integrating a motion model into the INR framework, presenting a new formulation for radial MRI reconstruction, and introducing a coarse-to-fine hash coding approach. Empirical evaluations on several MRI datasets demonstrate that our Moner achieves SOTA performance in terms of both efficiency and image quality.

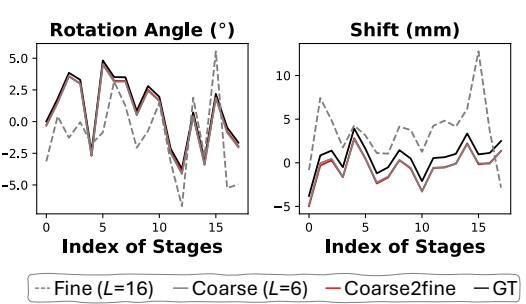

Figure 8: Motion trajectory by our Moner with different hash encodings on a sample (#5) of the fastMRI dataset for AF = 2× and MR = ±5.

**Limitation**  While the proposed Moner demonstrates promising MoCo performance, it has several limitations. First, our Moner is currently designed for 2D MRI, while 3D MRI MoCo is more practical, as subject's movements occur in 3D space. Secondly, the quasi-static motion model assumes rigid motion between acquisition frames, but it falls short in simulating certain complex motion scenarios. These include motion occurring within acquisition frames, non-rigid motion, and intricate spin-history effects. However, we emphasize that extending it to 3D is feasible with advanced INR architectures, such as $K$-plane (Fridovich-Keil et al., 2023) and Hexplane (Cao & Johnson, 2023). Also, for non-rigid motion, Moner can be extended by incorporating deformation field modeling (Reed et al., 2021). Additionally, our Moner primarily focuses on radial MRI by leveraging the naive Fourier-slice theorem. Adapting it to handle more diverse MRI sampling patterns (*e.g.*, Cartesian and spiral) also presents a potential direction for future work.

**Acknowlegement**  This study was supported by the National Natural Science Foundation of China (grant 62471296 & 62071299), the National Key R&D Program of China (grant 2024YFC2421100), and MoE Key Lab of Intelligent Perception and Human-Machine Collaboration (ShanghaiTech University). Also, the authors thank Bowen Li, Jie Feng, and Guoyan Lao for their help with 3D MRI data acquisition.

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

# A   APPENDIX

## A.1   AN EXTENSION OF OUR MONER FOR 3D RADIAL MRI

**Method**   Our Moner is currently designed for 2D radial MRI. Here, we demonstrate its extension to 3D radial MRI. Using a 3D stack-of-radial sampling (Feng, 2022), the 2D inverse Fourier transform of a spoke array from a specific view corresponds to the respective 2D projections, as illustrated in

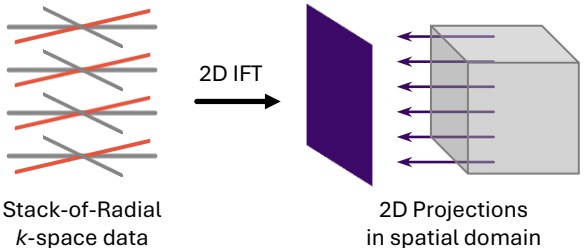

Figure 9: Illustration of the relationship between stack-of-radial $k$-space data and 2D projections in the spatial domain. The 2D inverse Fourier transform (IFT) of a spoke array (orange) from a specific view corresponds to the respective 2D projections (purple).

Fig. 9. This allows Moner to be extended to 3D radial MRI by solving a parallel back-projection problem in 3D space. Moreover, we learn 6 motion parameter $(\vartheta, \varphi, \psi, \tau_x, \tau_y, \tau_y)$ for each motion state to account for subject's movements occurring in 3D space. Specifically, a rotation matrix $\boldsymbol{A}(\vartheta, \varphi, \psi) \in \mathbb{R}^{3 \times 3}$ and a shift vector $\boldsymbol{\tau} \in \mathbb{R}^3$ are defined as below:

$$
\begin{aligned}
\boldsymbol{A}(\vartheta, \varphi, \psi) &= \begin{pmatrix} \cos\psi & -\sin\psi & 0 \\ \sin\psi & \cos\psi & 0 \\ 0 & 0 & 1 \end{pmatrix} \begin{pmatrix} \cos\varphi & 0 & \sin\varphi \\ 0 & 1 & 0 \\ -\sin\varphi & 0 & \cos\varphi \end{pmatrix} \begin{pmatrix} 1 & 0 & 0 \\ 0 & \cos\vartheta & -\sin\vartheta \\ 0 & \sin\vartheta & \cos\vartheta \end{pmatrix}, \\
\boldsymbol{\tau} &= \begin{pmatrix} \tau_x \\ \tau_y \\ \tau_z \end{pmatrix}.
\end{aligned}
\tag{10}
$$

By integrating the rotation matrix $\boldsymbol{A}(\vartheta, \varphi, \psi)$ and shift vector $\boldsymbol{\tau}$ into the spatial transformation (Eq. 5), our Moner can effectively model and correct the rigid motion in 3D space.

**Results on Simulated Data** To test the effectiveness of our Moner in 3D radial MRI, we conduct a simulation study on a 3D brain MR image with dimensions of $240 \times 240 \times 240$ acquired by a 3T Siemens MRI scanner. We use a 3D stack-of-radial sampling pattern with the golden-angle acquisition scheme. Detailed parameters are as follows: FOV = $283 \times 283 \times 283$, image spacing = $1 \times 1 \times 1$ mm$^3$, and total spoke views = 720. The AFs are set $2 \times$ and $4 \times$, corresponding to 360 and 180 views, respectively. We also simulate two level of rigid motion $\beta = \{5, 10\}$ with random rotations of $[-\beta, \beta]^\circ$ and shifts $[-\beta, \beta]$ mm along the X-axis, Y-axis and Z-axis. Note that here we consider 120 motion states, *i.e.*, our model totally estimates $120 \times 6 = 720$ motion parameters.

Fig. 10 presents the qualitative comparisons. NuIFFT struggles to achieve satisfactory MRI reconstructions, producing noticeable artifacts. In contrast, our method produces results that are visually close to the GTs, preserving both global structures and local details. Quantitative results in Table 6 further validate the significant improvements of our method over traditional NuIFFT. Additionally, Fig. 10 shows that our Moner accurately estimates 3D motion parameters.

Table 6: Quantitative results of 3D MR image by NuIFFT and our Moner the 3D brain data. Results of t-test statistical tests comparing our Moner to NuIFFT are denoted by ** ($p$-value $< 0.01$), * ($p$-value $< 0.05$), and ▼ (not significant, $p$-value $\geq 0.05$).

| AF | MR | NuIFFT | | Moner (Ours) | |
|---|---|---|---|---|---|
| | | PSNR | SSIM | PSNR | SSIM |
| $2\times$ | $\pm 5$ | 23.64** | 0.499** | **31.58** | **0.844** |
| | $\pm 10$ | 21.77** | 0.435** | **31.64** | **0.843** |
| $4\times$ | $\pm 5$ | 23.29** | 0.439** | **30.68** | **0.810** |
| | $\pm 10$ | 21.81** | 0.391** | **30.70** | **0.813** |

**Results on Real-World Data** To evaluate the effectiveness of our Moner on real-world MRI data, we collect two T1w brain 3D scans from a single subject using a 3.0 T United Imaging Healthcare (UIH) uMR 790 scanner. To obtain motion-free data and motion-corrupted data, the subject

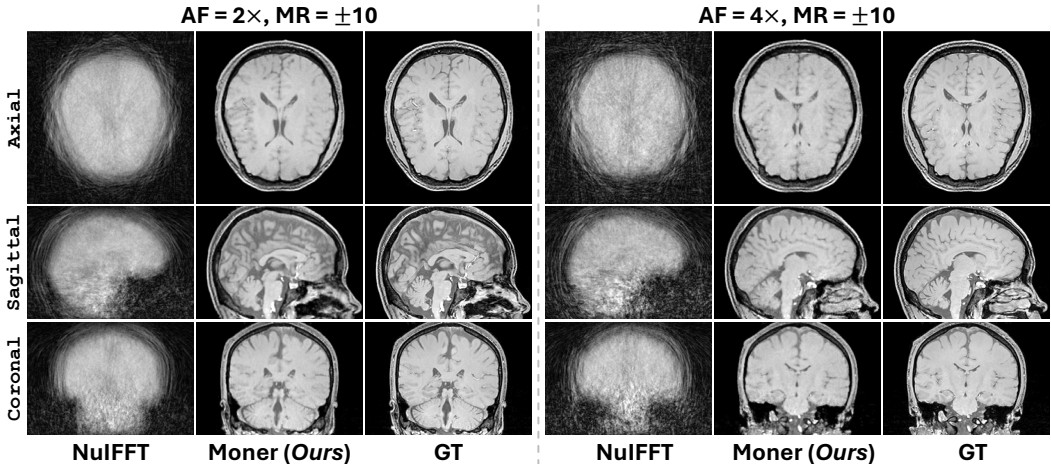

Figure 10: Qualitative results of 3D MR images (Axial, Sagittal, and Coronal views) by NuIFFT and our Moner on the 3D brain data for AF = 2×, MR = ± 10 and AF = 3×, MR = ± 10.

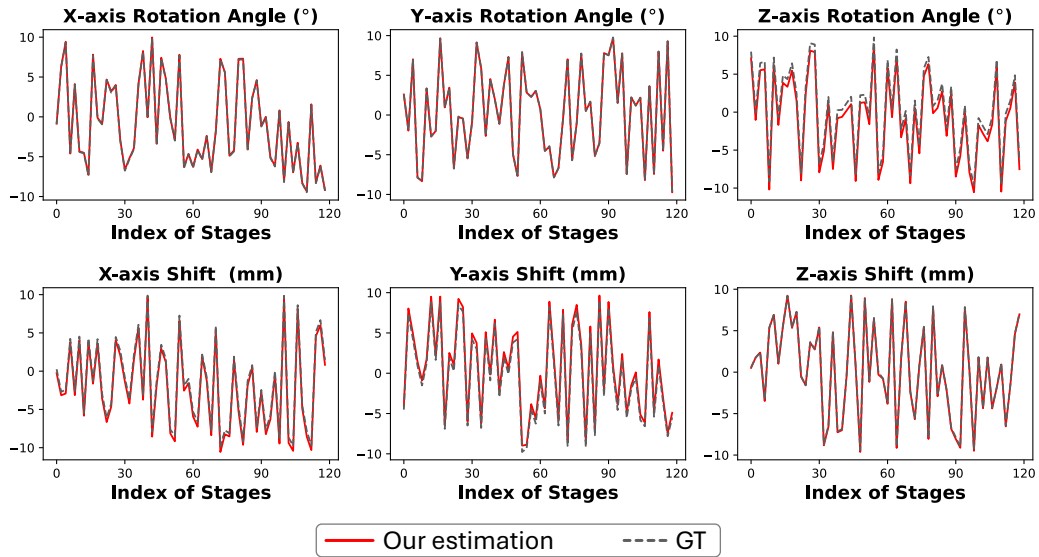

Figure 11: Qualitative results of motion parameter estimation by our Moner on the 3D brain data for AF = 4× and MR = ± 10.

is instructed to remain still or make abrupt movements (head shaking motion) three times during acquisition. The used radial Spoiled gradient echo sequence parameters we used are as follows: TR = 7.1 ms; TE = 3 ms; flip angle = 17°; matrix size = 120×120×120; FOV = 240×240×240 mm³; number of spokes = 65,000. Note that the data acquisition is approved by the institutional review board. Our Moner model does not access any prior motion knowledge. It instead assumes 650 motion stages and thus estimates unique *i.e.*, 650×6 = 3,900 motion parameters.

Fig. 12 shows the visualizations. Due to the 3D reconstruction task, only NuIFFT is included as an compared method. The result of NuIFFT suffers severely from motion artifacts, losing nearly a lots of image details. In contrast, our Moner's reconstruction closely resembles the motion-free reference in global structures and successfully recovers tissue details, such as the lateral ventricles, as highlighted in the zoomed-in view. Moreover, we shows the visualizations of motion trajectory estimated by our Moner in Fig. 13. The estimated trajectory is generally consistent with the motion pattern. In summary, this study preliminarily validates the effectiveness of our method on real-world MRI data.

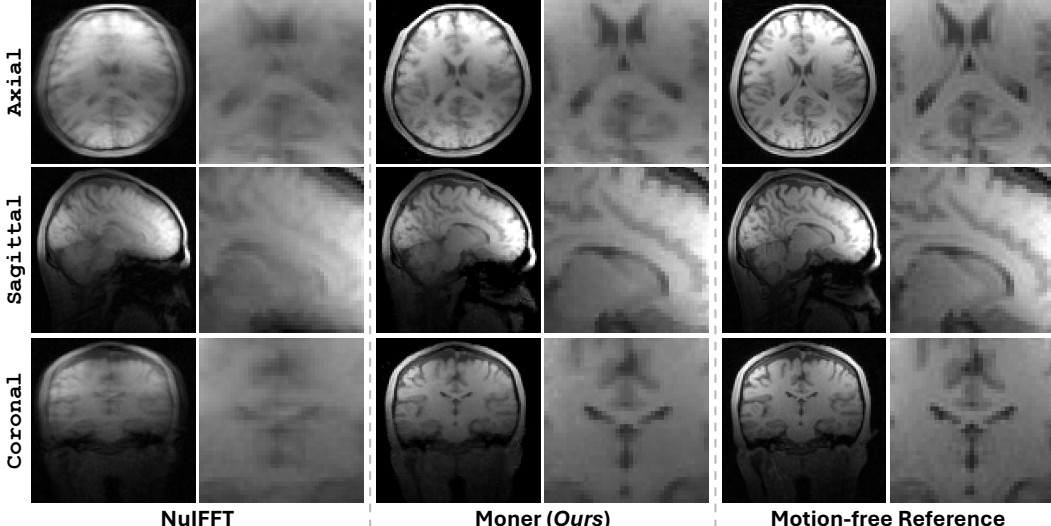

Figure 12: On a real-world T1w human brain 3D MRI volume with dimensions of $120{\times}120{\times}120$ and a voxel size of $2{\times}2{\times}2$ mm$^3$, acquired using a UIH uMR 790 scanner, our Moner effectively reduces motion artifacts and recovers both global structures and detailed image features.

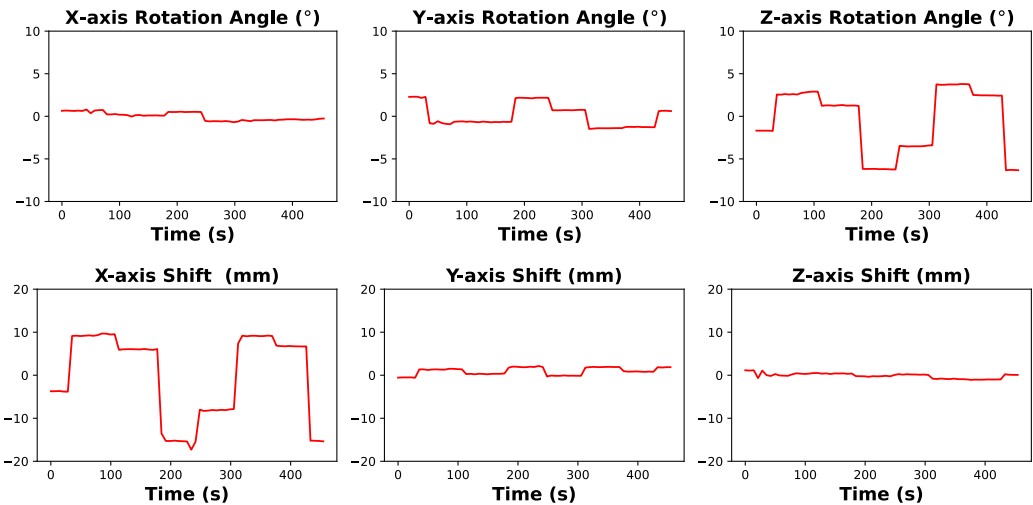

Figure 13: Visualizations of motion trajectory estimated by our Moner on a real-world T1w human bran 3D MRI volume with dimensions of $120{\times}120{\times}120$ and a voxel size of $2{\times}2{\times}2$ mm$^3$, acquired using a UIH uMR 790 scanner. Scanning a single spoke requires 7.1 ms. The whole acquisition (6,5000 spokes in total) thus takes about 461 s. The subject is instructed to make abrupt movements (head shaking motion) three times during the acquisition. The estimated trajectory is generally consistent with the motion pattern.

## A.2 ADDITIONAL DETAILS OF METRICS

**MR Image Quality**  In our evaluation, two commonly used visual metrics—PSNR and SSIM—are employed to assess the quality of reconstructed MR images. These metrics are implemented using the Python library skimage (**https://github.com/scikit-image/scikit-image**). However, since the GT and reconstructed images may not be at the same space and PSNR is a pixel-wise metric, we perform rigid registration to align the reconstructed MR images with the GT images before calculating these metrics. This registration is conducted using the Python library ants (**https://github.com/ANTsX/ANTs**).

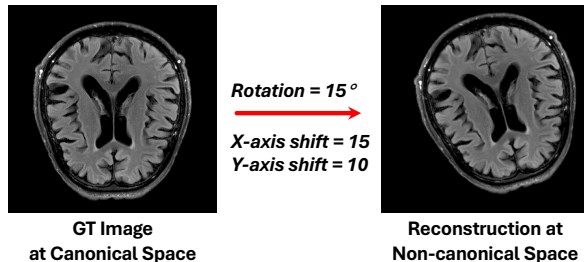

Figure 14: Illustration of the linear rigid transformation between GT at canonical space and reconstructed image at non-canonical space. Using the conventional $\ell_1$ metric (Eq. 11), the rotation and shift errors are $15°$ and 12.5, respectively. However, when applying our MoCo metrics (Eq. 12 and Eq. 13), both rotation and shift errors are measured to 0.

**MoCo Accuracy** To measure the MoCo accuracy, many works (Levac et al., 2023; 2024; Chen et al., 2023a) typically calculate the $\ell_1$ distance, defined by

$$\ell_1(\vartheta) = \frac{1}{n} \sum_{i=1}^{n} |\vartheta_i - \hat{\vartheta}_i|, \qquad \ell_1(\boldsymbol{\tau}) = \frac{1}{n} \sum_{i=1}^{n} |\boldsymbol{\tau}_i - \hat{\boldsymbol{\tau}}_i|, \tag{11}$$

where $\vartheta_i$ and $\boldsymbol{\tau}_i$ denote the GTs, while $\hat{\vartheta}_i$ and $\hat{\boldsymbol{\tau}}_i$ represent the predictions. However, the $\ell_1$ metric cannot accurately assess MoCo performance. In particular, a linear rigid transformation may exist between the GT image and the reconstructed image, meaning they are not aligned in the same space. While this linear transformation does not degrade image quality, it can lead to large $\ell_1$ errors. As shown in Fig. 14, the GT and reconstructed images are visually very similar, yet the $\ell_1$ MoCo errors are significantly high (rotation = $15°$, shift = 12.5). To more accurately evaluate MoCo performance, we propose calculating the standard deviation of the absolute errors between GTs and predictions, which is defined as below:

$$\sigma_\vartheta = \sqrt{\frac{1}{n} \sum_{i=1}^{n} (\Delta\vartheta_i - \mu_\vartheta)^2}, \quad \text{with} \quad \mu_\vartheta = \frac{1}{n} \sum_{i=1}^{n} \Delta\vartheta_i, \tag{12}$$

$$\sigma_{\boldsymbol{\tau}} = \sqrt{\frac{1}{n} \sum_{i=1}^{n} (\Delta\boldsymbol{\tau}_i - \mu_{\boldsymbol{\tau}})^2}, \quad \text{with} \quad \mu_{\boldsymbol{\tau}} = \frac{1}{n} \sum_{i=1}^{n} \Delta\boldsymbol{\tau}_i. \tag{13}$$

A lower value of the proposed metric denotes greater MoCo accuracy. The intuition behind this metric is that if the transformation between the GT and reconstructed images is linear, the errors in the predicted MoCo parameters across different spokes are identical, *i.e.*, $\Delta\vartheta_1 = \Delta\vartheta_2 = \cdots = \Delta\vartheta_n$ and $\Delta\boldsymbol{\tau}_1 = \Delta\boldsymbol{\tau}_2 = \cdots = \Delta\boldsymbol{\tau}_n$, resulting in a standard deviation of 0. Compared to the conventional $\ell_1$ metric, our new metric provides a more accurate assessment of MoCo accuracy.

### A.3 ADDITIONAL DETAILS OF BASELINES

We compare our Moner model with four representative MRI MoCo algorithms. Here, we provide the implementation details of these baselines to improve the reproducibility of this work.

**NuIFFT** Non-uniform inverse fast Fourier transform (NuIFFT) (Fessler, 2007) is an analytical reconstruction algorithm designed for MRI with non-uniform sampling patterns, such as radial and Poisson sampling. It first uses an interpolation algorithm (*e.g.*, linear) to generate uniform *k*-space data, followed by applying the IFFT operator to reconstruct the final MR images. We implement it using the function `KbNufftAdjoint` from the Python library `torchkbnufft` (**https://github.com/mmuckley/torchkbnufft**).

**TV** Total variation (TV) (Rudin et al., 1992) is a widely used explicit regularizer for various ill-posed inverse reconstruction problems. For the MRI MoCo problem, we solve the following opti-

mization problem:

$$\boldsymbol{f}^*, \boldsymbol{m}^* = \arg\min_{\boldsymbol{f}, \boldsymbol{m}} \frac{1}{2}\|\mathcal{T}\{\mathcal{R}(\boldsymbol{f}; \boldsymbol{m})\} - \boldsymbol{k}\|_2^2 + \lambda \cdot \text{TV}(\boldsymbol{f}), \tag{14}$$

where $\mathcal{T}\{\cdot\}$ is the Fourier transform, and $\mathcal{R}(\cdot)$ denotes the rigid transformation (rotation and shift) of the image $\boldsymbol{f}$ according to the motion parameter $\boldsymbol{m}$, with a weight $\lambda$ set to 1e-3. We solve the optimization problem using the PyTorch automatic differentiation framework, with a learning rate of 1e-3 and an optimization epoch of 200. The Adam algorithm (Kingma & Ba, 2014), with default settings, is employed. Here the hyper-parameters were tuned using 5 samples from the training set of the fastMRI dataset (Knoll et al., 2020) and were kept consistent across all other cases.

**DRN-DCMB** Deep residual network with densely connected multi-resolution blocks (DRN-DCMB) (Liu et al., 2020) is a supervised end-to-end DL model for the MRI MoCo reconstruction. It trains deep neural networks on paired MRI datasets to learn the inverse mapping from low-quality MR images to high-quality ones. Following the original paper, we implement and train DRN-DCMB using the training and validation sets from the fastMRI dataset (Knoll et al., 2020). It is important to note that an independent model is trained for each MRI acquisition setting, *i.e.*, we independently train 8 models for 8 cases (2 AFs × 4 MRs) in our experiments.

**Score-MoCo** Levac et al. (2023; 2024) proposed Score-MoCo, a SOTA approach for the rigid motion-corrupted MRI. This approach pre-trains a score-based generative model to provide high-quality prior images. During the inference phase, the model optimizes both the motion parameters and the underlying image, ultimately searching for reconstruction results that satisfy both data consistency and the distribution prior. We reproduce the results based on their official code (**https://github.com/utcsilab/motion_score_mri**) with appropriate modifications to match our experimental settings. As the official pre-trained model is trained on only T2 brain MRI images, we take the diffusion model trained on the fastMRI (Knoll et al., 2020) BRAIN data from Chung & Ye (2024). *The hyper-parameters were tuned using 5 samples from the training set of the fastMRI dataset (Knoll et al., 2020) and were kept consistent across all other cases.*

Table 7: Quantitative results (Mean±STD in SSIM) of MR images by compared methods on the fastMRI and MoDL datasets. Results of t-test statistical tests comparing our Moner to baselines are denoted by ** ($p$-value < 0.01), * ($p$-value < 0.05), and ▼ (not significant, $p$-value ≥ 0.05). Here the "AF" and "MR" represent acceleration rate and motion range, respectively. The best and second performances are highlighted in **bold** and underline, respectively.

| Dataset | AF | MR | Analy. | Optim. | Sup. | | Unsup. |
|---|---|---|---|---|---|---|---|
| | | | **NuIFFT** | **TV** | **DRN-DCMB** | **Score-MoCo** | **Moner (Ours)** |
| fastMRI | 2× | ±2 | 0.537±0.057** | 0.860±0.090* | 0.836±0.102** | 0.866±0.045** | **0.915±0.068** |
| | | ±5 | 0.364±0.065** | 0.859±0.090* | 0.716±0.104** | 0.849±0.038** | **0.914±0.067** |
| | | ±10 | 0.274±0.074** | 0.859±0.091* | 0.624±0.118** | 0.851±0.042** | **0.915±0.067** |
| | | ±15 | 0.251±0.074** | 0.859±0.091* | 0.532±0.102** | 0.857±0.041** | **0.913±0.071** |
| | 4× | ±2 | 0.460±0.042** | 0.613±0.202** | 0.812±0.096** | 0.813±0.052** | **0.888±0.077** |
| | | ±5 | 0.308±0.065** | 0.645±0.203** | 0.715±0.116** | 0.800±0.060** | **0.879±0.078** |
| | | ±10 | 0.221±0.067** | 0.639±0.210** | 0.570±0.124** | 0.809±0.055** | **0.882±0.081** |
| | | ±15 | 0.206±0.066** | 0.610±0.208** | 0.535±0.112** | 0.804±0.055** | **0.881±0.079** |
| MoDL | 2× | ±2 | 0.513±0.048** | 0.888±0.015** | 0.793±0.021** | 0.766±0.156** | **0.952±0.006** |
| | | ±5 | 0.336±0.041** | 0.887±0.016** | 0.613±0.019** | 0.716±0.185** | **0.952±0.006** |
| | | ±10 | 0.256±0.041** | 0.887±0.016** | 0.549±0.040** | 0.718±0.180** | **0.952±0.007** |
| | | ±15 | 0.246±0.041** | 0.886±0.015** | 0.356±0.029** | 0.795±0.043** | **0.949±0.010** |
| | 4× | ±2 | 0.420±0.040** | 0.861±0.063** | 0.696±0.033** | 0.552±0.158** | **0.938±0.011** |
| | | ±5 | 0.257±0.036** | 0.830±0.106** | 0.578±0.031** | 0.628±0.114** | **0.933±0.011** |
| | | ±10 | 0.197±0.035** | 0.835±0.100** | 0.382±0.035** | 0.643±0.182** | **0.932±0.013** |
| | | ±15 | 0.185±0.037** | 0.811±0.115** | 0.371±0.043** | 0.643±0.168** | **0.933±0.007** |

A.4 ADDITIONAL VISUAL RESULTS

Fig. 15, Fig. 16, Fig. 17, and Fig. 18 show additional reconstructed MR images. The proposed Moner method obtains the SOTA reconstructions.

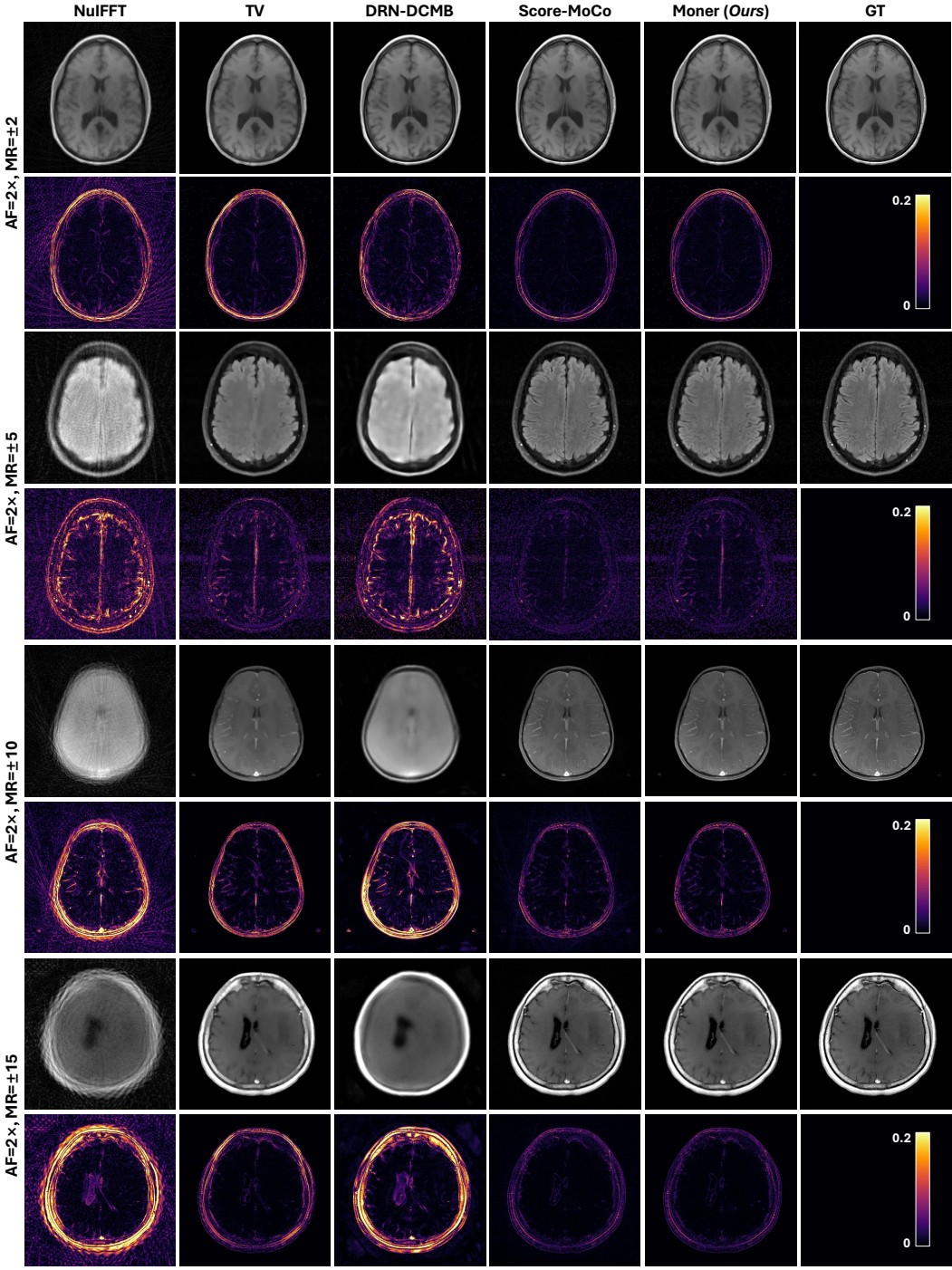

Figure 15: Qualitative results of MR images by compared methods on four test samples of the fastMRI for AF = $2\times$ and various MR (MR = $\pm2, \pm5, \pm10, \pm15$).

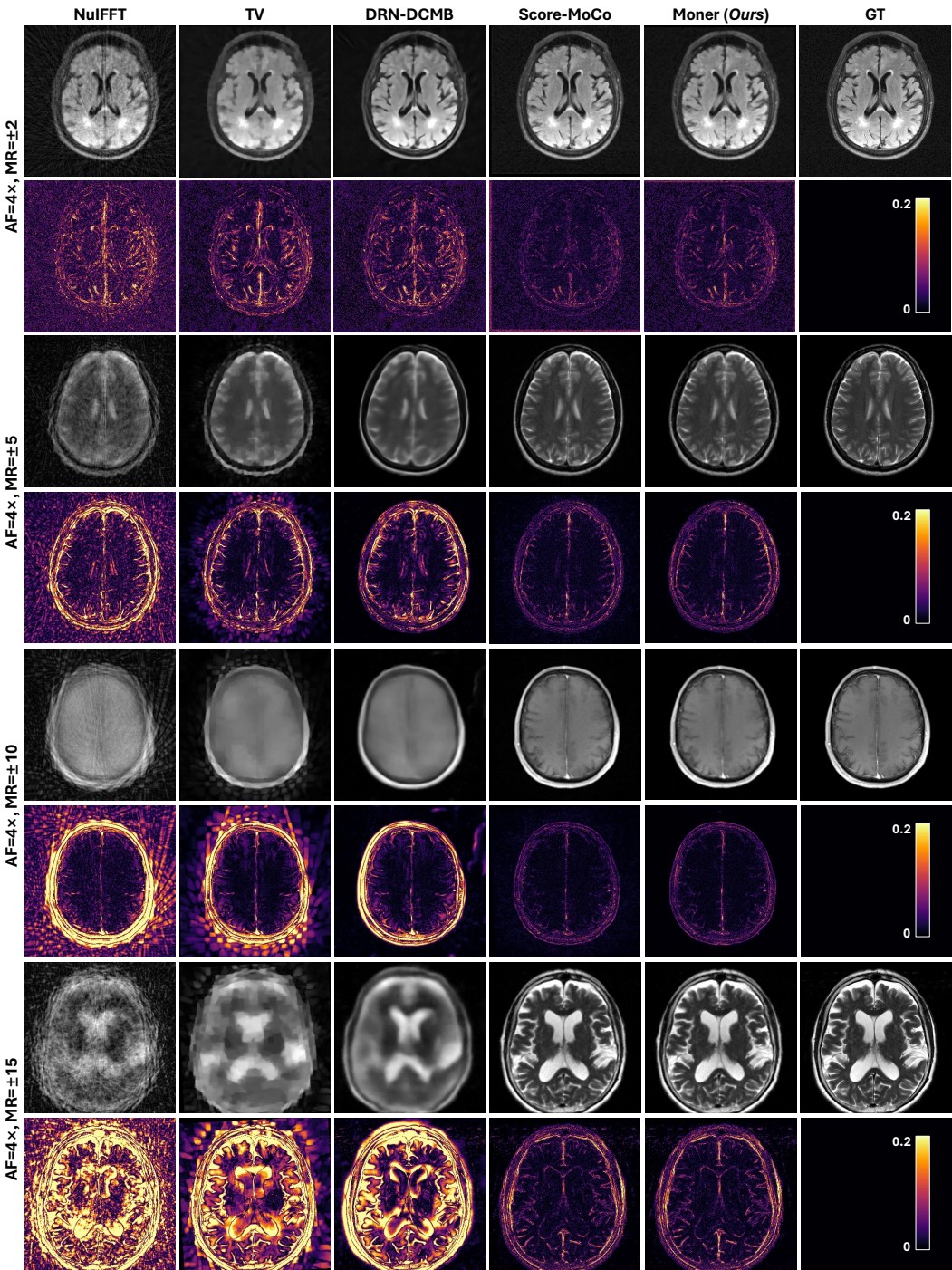

Figure 16: Qualitative results of MR images by compared methods on four test samples of the fastMRI for AF = 4× and various MR (MR = ±2, ±5, ±10, ±15).

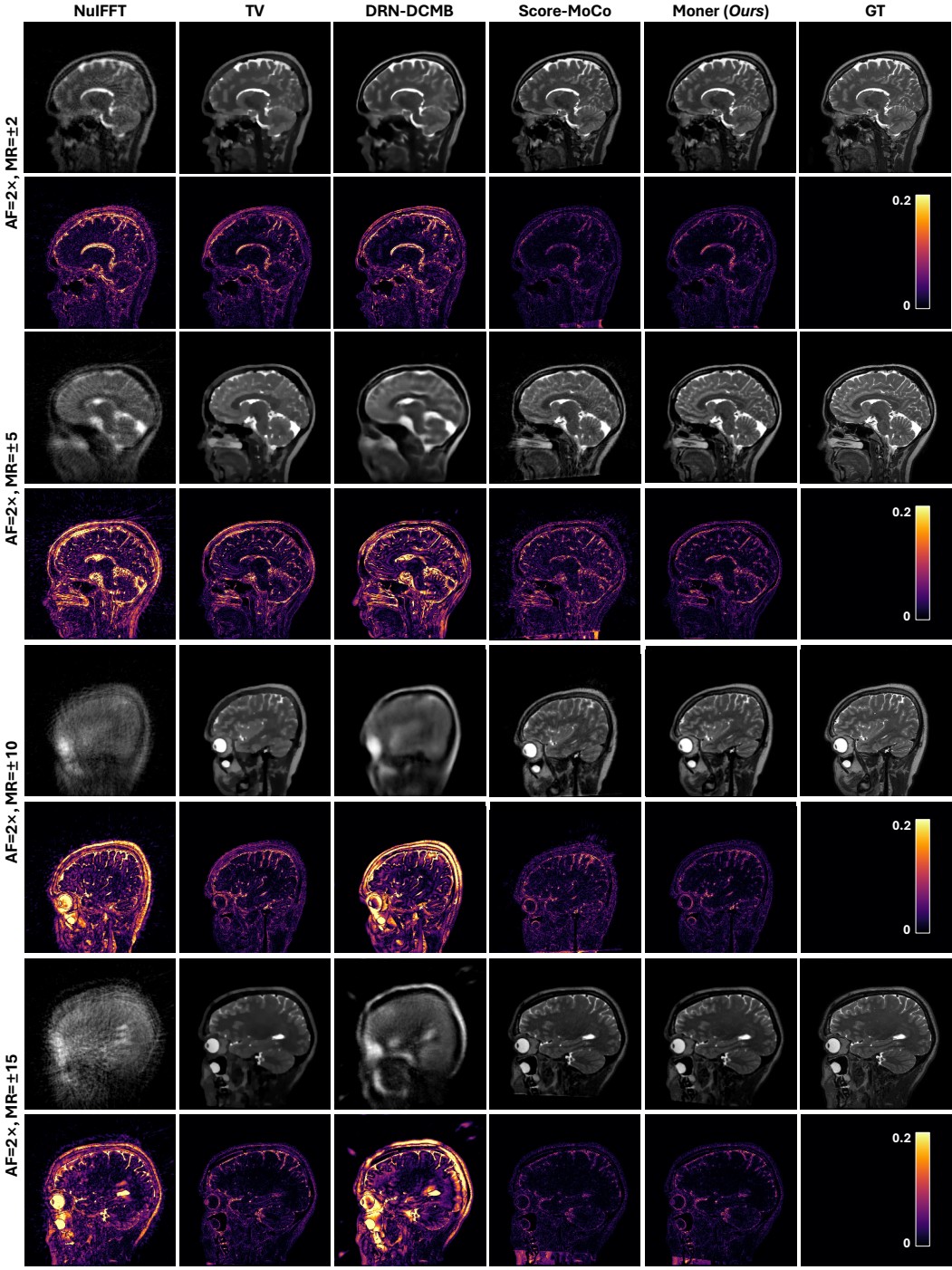

Figure 17: Qualitative results of MR images by compared methods on four test samples of the MoDL for AF = 2× and various MR (MR = ±2, ±5, ±10, ±15).

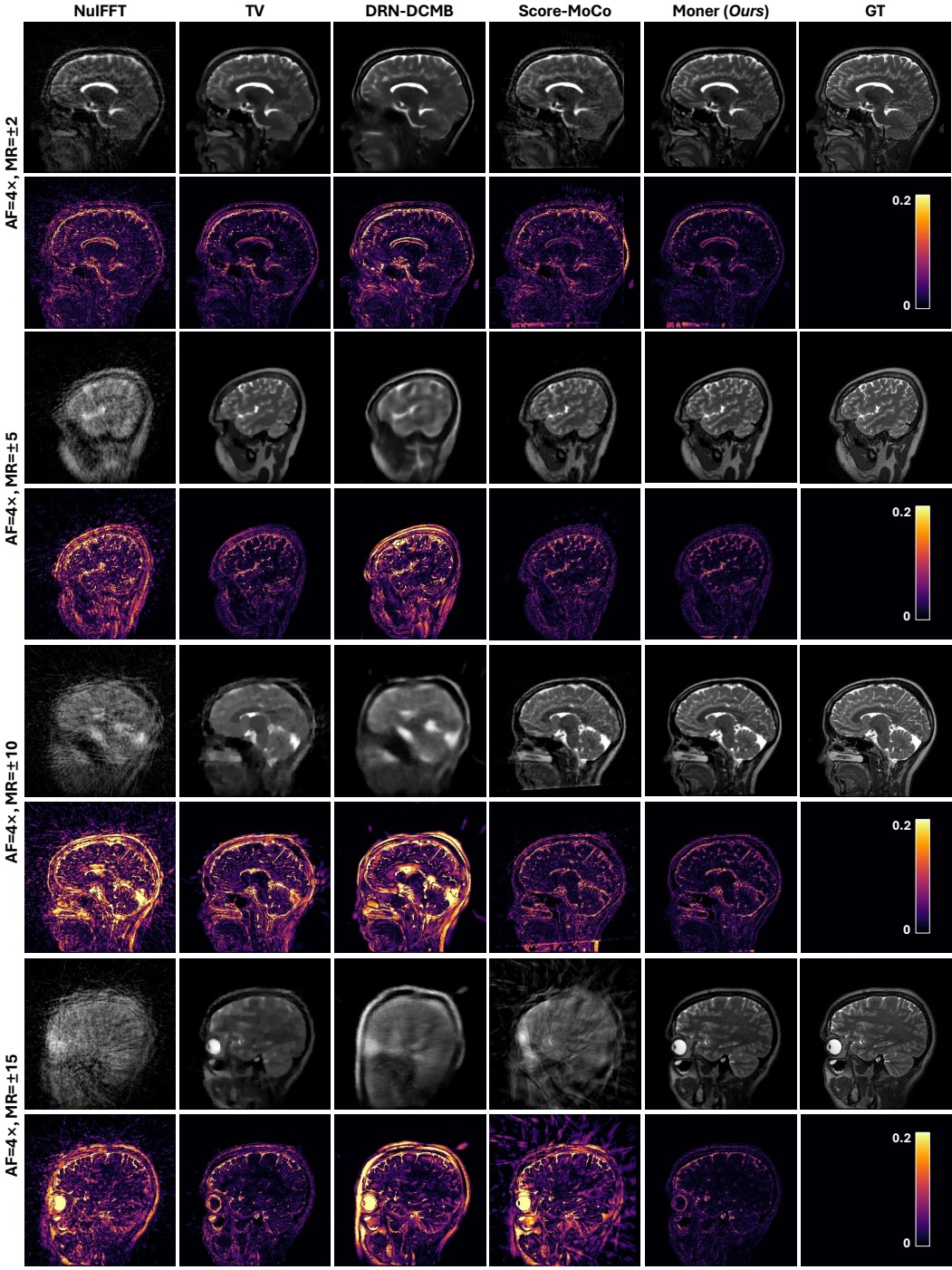

Figure 18: Qualitative results of MR images by compared methods on four test samples of the MoDL for AF = 4× and various MR (MR = ±2, ±5, ±10, ±15).

