# — SUPPLEMENTARY MATERIAL —

# MONER: MOTION CORRECTION IN UNDERSAMPLED RADIAL MRI WITH UNSUPERVISED NEURAL REPRESENTATION

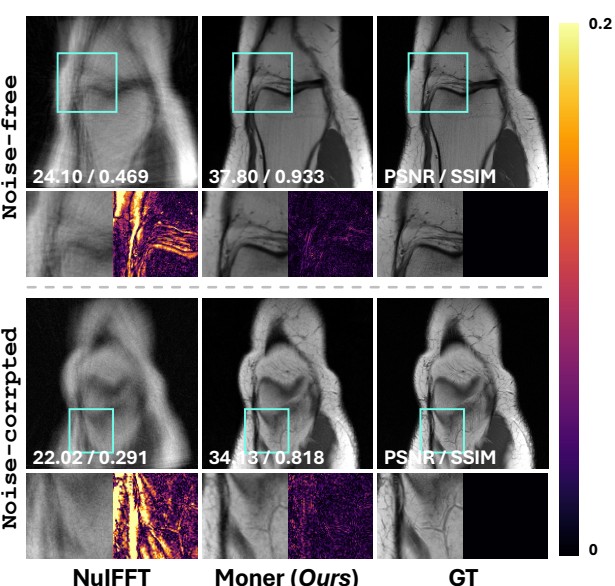

**Figure R 1.** Qualitative and quantitative results of MR images with and without noise by NuIFFT and our Moner on the fastMRI knee dataset (Knoll et al., 2020) for AF = 2×, MR = ± 10.

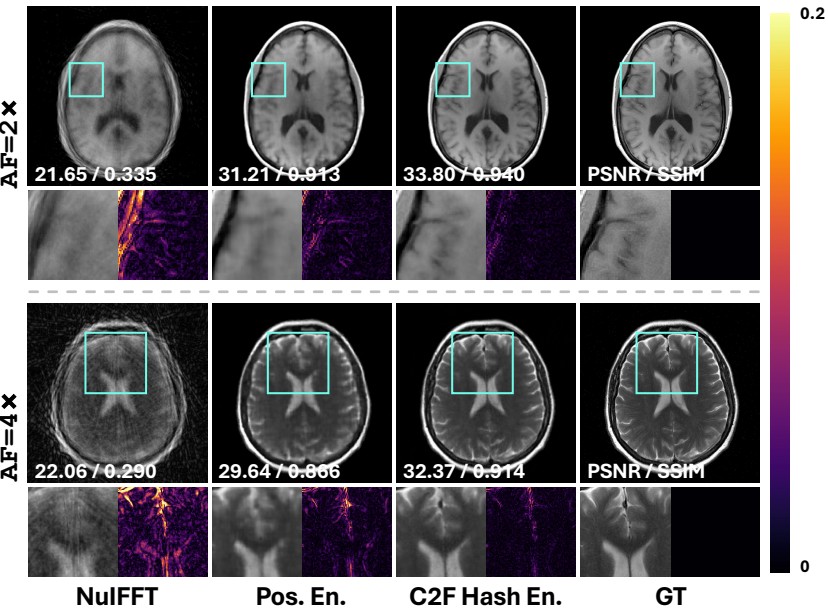

**Figure R 2.** Qualitative and quantitative results of MR images by our Moner with two different encoding modules on the fastMRI brain dataset (Knoll et al., 2020) for AF = $2\times$, MR = $\pm 10$.

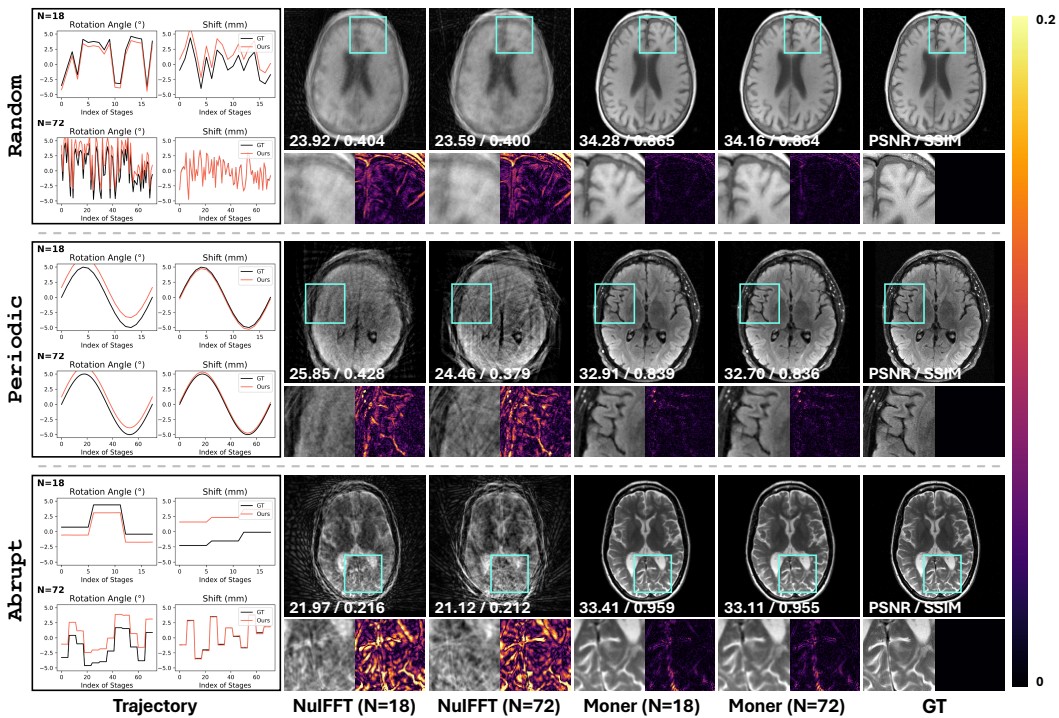

**Figure R 3.** Qualitative and quantitative results of our Moner when simulating different types of motion trajectories and different numbers of motion stages on the fastMRI brain dataset (Knoll et al., 2020) for AF = $2\times$, MR = $\pm 5$.

## REFERENCES

Florian Knoll, Jure Zbontar, Anuroop Sriram, Matthew J Muckley, Mary Bruno, Aaron Defazio, Marc Parente, Krzysztof J Geras, Joe Katsnelson, Hersh Chandarana, et al. fastmri: A publicly available raw k-space and dicom dataset of knee images for accelerated mr image reconstruction using machine learning. *Radiology: Artificial Intelligence*, 2(1):e190007, 2020.