# OpenReview forum: "Moner: Motion Correction in Undersampled Radial MRI with Unsupervised Neural Representation"
_ICLR.cc/2025/Conference — ICLR 2025 Spotlight_

### Official Review · Reviewer_kTjY · 2024-11-02

**Soundness:** 3
**Presentation:** 3
**Contribution:** 3
**Rating:** 8
**Confidence:** 5

**Summary:**

The paper proposes to use implicit neural representations (INRs) to perform unsupervised joint reconstruction and motion estimation from undersampled and motion corrupted 2D radial MRI measurements.

The method achieves the best motion estimation results across datasets compared to classical and supervised deep learning-based methods for motion estimation.
In terms of reconstructed image quality, the method falls behind supervised deep learning for in-distribution performance but exhibits much more robustness against out-of-distribution examples.
In terms of computation time the method is 5-6 times faster than the diffusion model baseline requiring 3-5 minutes per 2D slice.

**Strengths:**

1.	(Originality) It is the first paper that explores the use of untrained neural networks (here in the form of INRs) for joint reconstruction and motion estimation. The proposed extensions in terms of an adapted hash encoding optimization schedule and a loss function in the projection space are interesting.
2.	(Significance) The investigated problem of motion correction in (radial) MRI is important. The lack of training data from all possible domains required for the supervised baselines and the resulting lack of out-of-distribution robustness is a real problem and methods that are more robust are important.
3.	(Quality/clarity) Overall the paper is well written, and the experiments seem to be of good quality. However, some additional information and explanations are required in my opinion (see weaknesses). If those can be addressed adequately, I'm willing to raise the score.

**Weaknesses:**

1.	It is not clear how the motion trajectories, e.g., in Figure 7 are simulated – more information should be provided. Are there fixed intervals after which motion changes direction? In between those events does every spoke has its own motion state meaning that 300 spokes result intro estimating 300*3=900 unknowns? How does this model change when the number of spokes changes due to a different acceleration factor?
2.	Section 4.1 does not contain enough information about the dataset used in the experiments. More information should be provided either in the main body or the appendix. 25 test slices is not much depending on how the data is selected. Are the 25 slices the entire slices from e.g. two subjects or are they 25 mid-slices from 25 subjects? The fastMRI dataset is very diverse. Are the subjects for training and testing taken from the same contrasts, e.g., T2 weighted or FLAIR? What is the main difference of the MoDL dataset to the data used from the fastMRI dataset? This information is important to assess the severity of the distribution shift. From Figure 4 it seems that the main difference is the view being axial and coronal respectively. Are there any other differences?
3. The comparison of the computational speed would benefit from a more detailed discussion. Where do the differences in terms of speed come from? How can the TV method with 200 epochs be slower than the INR with 4000 epochs? What is responsible for the speed difference compared to the diffusion model – the size of the networks or the number of steps? (What is the size of the MLP that is used?)

**Questions:**

1.	Is formulating the loss in the projection space specific to the radial sampling trajectory or can it also be used for e.g. Cartesian sampling as well?
2. It would be helpful to add scores to the qualitative comparisons in e.g. Figure 4 to provide a better feeling how differences in the scores translate to perceivable differences.
3. Why is the diffusion model baseline categorized as self-supervised e.g. in line 309? As the model requires fully sampled data for training I would consider it as a supervised baseline.
4. How many spokes are acquired for the acceleration factors of 2 and 4 respectively?

---

> ### Author Response · Authors · 2024-11-20
>
> Thank you for the valuable comments. We are encouraged by your positive feedback. Below, we provide detailed point-by-point responses to address your concerns.
>
> ---
>
> **W1: It is not clear how the motion trajectories, e.g., in Figure 7 are simulated – more information should be provided. Are there fixed intervals after which motion changes direction? In between those events does every spoke has its own motion state meaning that 300 spokes result intro estimating 300*3=900 unknowns? How does this model change when the number of spokes changes due to a different acceleration factor?**
>
> **A:** Our motion simulation is based on the concept of motion stages [1][2]. Specifically, all spokes are divided into $N$ motion stages, where spokes in the same motion stage share identical motion trajectories. Our Moner model assigns 3 motion parameters per stage and thus totally estimates $N\times$3 unknowns. In our experiment, we use 360 and 180 spokes for acceleration factors (AFs) of 2$\times$ and 4$\times$, respectively. These spokes are divided into $N=18$ motion stages, resulting in the estimation of $18×3=54$ unknowns. We have added this information in `the revised submission (Sec. 4.1)`.
>
> Additionally, we evaluate our method across different numbers of motion stages ($N=\{18, 72\}$), corresponding to $18\times 3=54, 72\times 3=216$ unknowns, and simulate different types of motion trajectories (including random, periodic, and abrupt). Note that all hyperparameters for our model remained consistent with those reported in the original submission.
>
> As shown in Table R1, our Moner achieves excellent and stable reconstruction performance, indicating minor variations in PSNR ($\pm1$ dB) and SSIM ($\pm$0.2), and statistical tests show no significant differences. Qualitative results are provided in `Figure R3 of the Supplementary Material`, where reconstructed MR images across all conditions appear visually consistent, further confirming the robustness of our method.
>
>
> | Type of Motion   | Motion Stage  | PSNR           | SSIM           |
> |:----------------:|:----------------:|:----------------:|:----------------:|
> |       Random       |         N=18         |  32.71±1.03      |   0.907±0.046    |
> |       Random       |         N=72         |  32.48±1.09 ▼    |   0.904±0.048 ▼  |
> |      Periodic      |         N=18         |  32.88±1.11 ▼    |   0.910±0.048 ▼  |
> |      Periodic      |         N=72         |  32.45±0.78 ▼    |   0.906±0.047 ▼  |
> |       Abrupt       |         N=18         |  32.54±1.40 ▼    |   0.897±0.051 ▼  |
> |       Abrupt       |         N=72         |  31.75±1.08 ▼    |   0.884±0.058 ▼  |
>
>
> *Table R1: Quantitative results of our Moner on the fastMRI dataset for AF=2x and MR=±5 when simulating different types of motion trajectories and different numbers of motion stages. Results of t-test based statistical tests comoaring Row #1 (Random & N=18) to other settings are presented by \*\*(p<0.01), \*(p<0.05), and ▼ (no significant, p≥0.05).*
>
> > [1] Spieker, Veronika, et al. "Deep learning for retrospective motion correction in MRI: a comprehensive review." IEEE Transactions on Medical Imaging (2023).
>
> > [2] Levac, Brett, et al. "Accelerated motion correction with deep generative diffusion models." Magnetic Resonance in Medicine 92.2 (2024): 853-868.
>
> ---
>
> **W2: Section 4.1 does not contain enough information about the dataset used in the experiments. More information should be provided either in the main body or the appendix. 25 test slices is not much depending on how the data is selected. Are the 25 slices the entire slices from e.g. two subjects or are they 25 mid-slices from 25 subjects? The fastMRI dataset is very diverse. Are the subjects for training and testing taken from the same contrasts, e.g., T2 weighted or FLAIR? What is the main difference of the MoDL dataset to the data used from the fastMRI dataset?**
>
> **A:** In our experiment, the training, validation, and test sets of the fastMRI dataset are randomly selected from different subjects along the axial direction to ensure no data leakage between sets. These sets include a diverse combination of three contrasts: T1w, T2w, and FLAIR images. For the MoDL dataset, we used 20 T2w brain MR slices from 20 distinct subjects along the sagittal direction. This difference in orientation (axial vs. sagittal) and the exclusive use of T2w images introduce a distribution shift. We thus believe that, despite the limited number, these slices sufficiently represent the data diversity. We have added the dataset information in `the revised submission (Sec. 4.1)` to improve clarity. Additionally, we have a plan to evaluate the model on additional slices to further demonstrate the model generalization.

---

> ### Author Response · Authors · 2024-11-20
>
> **W3: The comparison of the computational speed would benefit from a more detailed discussion. Where do the differences in terms of speed come from? How can the TV method with 200 epochs be slower than the INR with 4000 epochs? What is responsible for the speed difference compared to the diffusion model – the size of the networks or the number of steps? (What is the size of the MLP that is used?)**
>
> **A:** The computational efficiency of our Moner model mainly benefits from two factors:
>
> - `Hash Encoding for Compact and Efficient Representations`: Hash encoding [3][4] explicitly learns features for each position in the imaging space, significantly enhancing the ability of the subsequent tiny MLP network (a two-layer structure) to fit high-frequency image details. This design accelerates convergence and reduces the computational overhead typically associated with training larger networks for high-resolution image reconstruction.
>
> - `Ray Tracing-Based Optimization:` Our method employs a ray tracing-based optimization pipeline. Specifically, for each projection $\boldsymbol{g}(\theta_i,\rho)$, multiple coordinates along the corresponding ray $\boldsymbol{r}(\theta_i,\rho)$ are sampled and input into the INR network. In each optimization epoch, our Moner only computes forward and backward passes for these sampled positions, rather than over the entire imaging space (i.e., FOV) as in TV and Score-MoCo. Consequently, TV with 200 epochs is slower than our Moner with 4000 epochs. Additionally, Score-MoCo relies on a deep U-Net or Transformer to iteratively denoise the noisy image, and backward optimization for both motion parameters and MR images is performed in each iteration. These factors significantly slow down its optimization.
>
>
> > [3] Müller, Thomas, et al. "Instant neural graphics primitives with a multiresolution hash encoding." ACM transactions on graphics (TOG) 41.4 (2022): 1-15.
>
> > [4] Barron, Jonathan T., et al. "Zip-nerf: Anti-aliased grid-based neural radiance fields." Proceedings of the IEEE/CVF International Conference on Computer Vision. 2023.
>
> ---
>
> **Q1: Is formulating the loss in the projection space specific to the radial sampling trajectory or can it also be used for e.g. Cartesian sampling as well?**
>
> **A**: Radial sampling benefits from each spoke passing through the center of  Fourier frequency space, where the 1D inverse Fourier transform of a spoke corresponds to an integral projection. In contrast, Cartesian sampling acquires k-space data on a regular grid, with each spoke parallel to the readout axis (i.e., k_x) and adjacent spokes separated by a fixed interval along the phase encoding axis (i.e., k_y). According to Fourier transform theory, translating a signal in k-space induces a phase shift in the spatial domain.
>
> Technically, *to extend our Moner model to Cartesian sampling, we propose applying a phase shift to the network-predicted MR image before integrating it into the projection forward model*. However, Cartesian sampling lacks the redundant low-frequency information inherent in radial sampling (i.e., central k-space coverage by spokes), which could decrease optimization stability. Therefore, Cartesian sampling with specifically designed trajectories to simultaneously scan high-frequency and low-frequency spokes [5][6] may be required. Moreover, adopting an alternating optimization strategy could help enhance model convergence and performance under Cartesian sampling patterns.
>
> > [5] Levac, Brett, Ajil Jalal, and Jonathan I. Tamir. "Accelerated motion correction for MRI using score-based generative models." 2023 IEEE 20th International Symposium on Biomedical Imaging (ISBI). IEEE, 2023.
>
> > [6] Levac, Brett, et al. "Accelerated motion correction with deep generative diffusion models." Magnetic Resonance in Medicine 92.2 (2024): 853-868.

---

> ### Author Response · Authors · 2024-11-20
>
> **Q2: It would be helpful to add scores to the qualitative comparisons in e.g. Figure 4 to provide a better feeling how differences in the scores translate to perceivable differences.**
>
> **A**: Thanks for your constructive suggestions. We have included the quantitative metrics (PSNR/SSIM) in `Figures 4, 5, 6, and 8 of the revised submission.`
>
> ---
>
> **Q3: Why is the diffusion model baseline categorized as self-supervised e.g. in line 309? As the model requires fully sampled data for training I would consider it as a supervised baseline.**
>
> **A:** Thank you for pointing this out. Unlike the traditional end-to-end learning paradigm, which relies on extensive paired MRI data, diffusion models can effectively capture data distributions using only numerous high-quality MR images. We thus categorize the diffusion-based Score-MoCo as a self-supervised model. However, we also agree with the reviewer that the diffusion model should be classified as a supervised baseline due to its heavy reliance on fully sampled MRI data. In the revised submission, we modify the categorization of Score-MoCo to supervised.
>
> ---
>
> **Q4: How many spokes are acquired for the acceleration factors of 2 and 4 respectively?**
>
> **A:** In our experimental setup, each spoke has a length of 511, corresponding to an imaging FOV of 511$\times$511. The fully sampled radial k-space data consists of approximately 720 views. Therefore, for acceleration factors (AFs) of 2 and 4, the number of spokes is 360 and 180, respectively. This information has been included in `the revised submission (Sec. 4.1)` to improve clarity and reproducibility.

---

> ### Comment · Reviewer_kTjY · 2024-11-25
> **Response to Official Comment by Authors**
>
> Thank you for the insigthful rebuttal. The additional experiments and information provided look good. I raised my score accordingly.

---

> > ### Author Response · Authors · 2024-11-26
> >
> > Thank you for your supporting our work! Your insightful comments have significantly enhanced it. Once again, we sincerely appreciate your efforts!

---

### Official Review · Reviewer_mfmU · 2024-11-04

**Soundness:** 3
**Presentation:** 4
**Contribution:** 3
**Rating:** 8
**Confidence:** 3

**Summary:**

This paper proposed an unsupervised motion correction method for undersampled radial MRI reconstruction, which fits implicit neural representation (INR) of the image to the observation (undersampled k-space with motion) at test time. MRI reconstruction with motion correction is a challenging task. Though applying INR to medical image reconstruction (e.g. MRI and CT) is not new, its application to motion correction is underexplored. This paper presents its great potential in solving the problem.

**Strengths:**

- The paper is overall well-written, presenting an interesting method valuable to the field.
- A good background introduction that let readers know what radial MRI reconstruction is.
- Methods were well represented, with motivation explained for each design choice.
- Extensive experiments and ablation studies prove the efficacy of the proposed method and individual design choices.

**Weaknesses:**

- Experiments were only conducted on brain MRI datasets with similar contrast/MR sequence. Readers would be interested in how good the proposed method is on MRI images of different contrasts or taken at different body parts. Specifically, will the optimization still converge stably with the same optimization strategy (e.g. hash encoding) and hyperparameter choice? In addition, how does noise level affect the optimization process?
- How will trans-plane motion affect the stability of optimization?
- In Fig. 8, I notice that the reconstructed image looks blurry and lacks some thin structures like vessels. In other figures, images are too small to observe such details. I suggest authors show zoomed-in images from both baselines and proposed method for better comparison.
- Authors may want to report FOV and pixel spacing to let readers get a sense of the magnitude of motion ranges used in this paper.
- It is surprising to see that a supervised method (DRN-DCMB) is worse than an unsupervised method. In Fig. 4, DRN-DCMB results look very blurry, making me doubt if it is the SOTA supervised method. Authors may consider alternative baselines such as [1].

[1] Singh NM, Iglesias JE, Adalsteinsson E, Dalca AV, Golland P. Joint frequency and image space learning for MRI reconstruction and analysis. The journal of machine learning for biomedical imaging. 2022 Jun;2022.

**Questions:**

- How were data from different coils handled in this paper? Were data from all coils fed into the model simultaneously during the optimization?

---

> ### Author Response · Authors · 2024-11-20
>
> Thank you for taking the time to review our work. We greatly appreciate your positive feedback. Below, we provide detailed point-by-point responses to address your concerns.
>
> ---
>
> **W1: Experiments were only conducted on brain MRI datasets with similar contrast/MR sequence. Readers would be interested in how good the proposed method is on MRI images of different contrasts or taken at different body parts. Specifically, will the optimization still converge stably with the same optimization strategy (e.g. hash encoding) and hyperparameter choice? In addition, how does noise level affect the optimization process?**
>
> **A:** Thank you for the insightful comment. In our experiments, the fastMRI dataset includes diverse contrasts, such as T1w, T2w, and FLAIR images, as shown in Figures 4 and 13. To improve clarity, this information has been explicitly added to the revised manuscript (Sec. 4.1).
>
> We agree that experiments using different body parts are essential for assessing model generalization. To address this, we additionally test our method on 10 knee T2w MR images from the fastMRI dataset. Gaussian noises (SNR=40 dB) are also simulated in the k-space data. Importantly, all hyperparameters and optimization strategies, including Coarse2fine hash encoding, were consistent with those used in our original experiments.
>
> As shown in Table R1, our Moner achieves consistent reconstruction quality on this new dataset, confirming its robustness. Qualitative results are provided in `Figure R1 of the Supplementary Material`, further supporting the stability and effectiveness of our method.
>
>
> | **Condition**     | **NuIFFT**                 | **Ours**                 |
> |:-----------------:|:-------------------------:|:------------------------:|
> | Noise-free        | 19.92±1.08** / 0.357±0.023** | 34.49±0.47 / 0.906±0.008 |
> | Noise-corrupted   | 19.57±1.07** / 0.239±0.011** | 32.46±0.44 / 0.776±0.018 |
>
> *Table R1: Quantitative results (PSNR/SSIM) of NuIFFT and our Moner on 10 knee samples of the fastMRI dataset for MR=±10 and AF=2x. Results of t-test based statistical tests comparing our Model to NuIFFT are presented by \*\* (p<0.01), \*(p<0.05), and ▼ (no significant, p≥0.05).*
>
> ---
>
> **W2: How will trans-plane motion affect the stability of optimization?**
>
> **A:** Thank you for the question. Our current Moner model is designed for 2D MRI motion correction (MoCo) and does not account for trans-plane motion in 3D space. To address this limitation, we extended Moner to a 3D version for 3D MRI MoCo. This extension explicitly models motion across all three spatial dimensions. Experimental results, provided in `the revised submission (Sec. A.1)`, demonstrate that 3D Moner achieves consistent and accurate motion-corrected reconstructions, highlighting its robustness in handling trans-plane motion.
>
> ---
>
> **W3: In Fig. 8, I notice that the reconstructed image looks blurry and lacks some thin structures like vessels. In other figures, images are too small to observe such details. I suggest authors show zoomed-in images from both baselines and the proposed method for better comparison.**
>
> **A:** Thanks for your constructive suggestions. We have included zoomed-in images for all comparison methods in Figure 4 to provide a more detailed comparison. In Figure 8, our focus is to show the influence of the Coarse2fine hash encoding strategy on the performance of our model. There are two main reasons for the ‘looks blurry’ mentioned by the reviewers:
>
> - Our experimental setup involves undersampling (AF = 2$\times$ in Figure 8), so the proposed unsupervised method, Moner, faces challenges not only from motion artifacts but also from the undersampling issue.
>
> - Since the reconstructed images and GTs may not be in the same space, we perform the rigid registration to align the reconstructions to GTs before calculating quantitative metrics. While the registration involves interpolation operators (e.g., linear interpolation) and thus may slightly cause blurring effects.
>
> ---
>
> **W4: Authors may want to report FOV and pixel spacing to let readers get a sense of the magnitude of motion ranges used in this paper.**
>
> **A:** In our experiment, the imaging FOV is 511$\times$511 mm$^2$, and the pixel spacing is 1$\times$1 mm$^2$. This information has been added to `the revised submission (Sec. 4.1)` to improve clarity and provide context for the motion ranges.

---

> > ### Author Response · Authors · 2024-11-20
> >
> > **W5: It is surprising to see that a supervised method (DRN-DCMB) is worse than an unsupervised method. In Fig. 4, DRN-DCMB results look very blurry, making me doubt if it is the SOTA supervised method. Authors may consider alternative baselines such as [1]**
> >
> > **A5:** Thank you for the insightful comment. Unlike traditional low-level vision tasks (e.g., image denoising), MRI MoCo presents unique challenges due to the highly diverse motion artifacts caused by varying movement patterns. The supervised DRN-DCMB method,  which relies on an end-to-end learning paradigm, lacks explicit modeling of motion, leading to suboptimal performance and the loss of fine image details, as reflected in the blurry results in Figure 4. We appreciate the suggestion to include Singh et al. [1], which corrects motion artifacts jointly in frequency and image space. This approach may provide a more robust supervised baseline. In the revised manuscript, we will incorporate this method as an additional comparison.
> >
> > However, it is worth noting that Score-MoCo, which combines pre-trained diffusion models with model-based optimization, could be considered the SOTA for MRI MoCo. Our results demonstrate that Score-MoCo offers superior reconstruction and slightly outperforms our Moner in in-domain scenarios. However, its performance degrades in out-of-domain settings, particularly for 4x acceleration MoCo tasks. In contrast, our Moner demonstrates superior robustness and generalization in such challenging scenarios.
> >
> > > [1] Singh NM, Iglesias JE, Adalsteinsson E, Dalca AV, Golland P. Joint frequency and image space learning for MRI reconstruction and analysis. The journal of machine learning for biomedical imaging. 2022 Jun;2022.
> >
> > ---
> >
> > **Q1: How were data from different coils handled in this paper? Were data from all coils fed into the model simultaneously during the optimization?**
> >
> > **A:** In our pipeline, after the MR image is predicted by the INR network, sensitivity maps are applied to generate multiple MR images corresponding to different coils. These coil-specific MR images are then transformed into projection data using the projection forward model. Finally, the INR network and motion parameters are optimized by minimizing the error between the predicted and real projection data across all coils.

---

> > > ### Comment · Reviewer_mfmU · 2024-11-25
> > >
> > > I appreciate the authors' response. All my concerns are addressed. As a result, I have updated my score to 8.

---

> > > > ### Author Response · Authors · 2024-11-26
> > > >
> > > > We are glad to know that your concerns have been addressed.  Your constructive comments have greatly improved our work. Thank you for your time and effort!

---

### Official Review · Reviewer_63Er · 2024-11-04

**Soundness:** 3
**Presentation:** 3
**Contribution:** 3
**Rating:** 6
**Confidence:** 3

**Summary:**

The paper proposes Moner, an unsupervised neural representation framework for motion correction (MoCo) in undersampled radial MRI, utilizing implicit neural representations (INR) to model motion without pre-training on high-quality MR datasets. Key innovations include a quasi-static motion model integrated into INR, reformulating MRI recovery as a back-projection problem via the Fourier-slice theorem, and a coarse-to-fine hash encoding strategy. Extensive experiments on fastMRI and MoDL datasets show that Moner not only achieves high-quality reconstructions but also demonstrates robustness across diverse acquisition conditions.

**Strengths:**

* Unsupervised Approach: Unlike many MoCo methods that require large pre-trained datasets, Moner’s unsupervised framework enhances generalizability and applicability across different MRI modalities.
* Motion Estimation and Stability: The quasi-static motion model and back-projection formulation stabilize model optimization, effectively reducing artifacts even in challenging motion-corrupted scenarios.
* Efficiency and Robustness: Compared to existing methods like Score-MoCo, Moner achieves superior out-of-domain (OOD) performance and faster reconstruction speeds, making it highly efficient and scalable.

**Weaknesses:**

* Extension to 3D and Non-Radial Sampling Patterns: The current model primarily addresses 2D radial MRI, and while extensibility to 3D MRI is suggested, it remains untested. The same applies to Cartesian or spiral sampling patterns, which are common in clinical MRI.
* Reliance on Motion Assumptions: The quasi-static motion model assumes rigid motion between acquisition frames. Further clarification on its limitations in more complex motion settings could strengthen the study.

**Questions:**

1. Could the authors elaborate on potential strategies to adapt Moner for 3D MRI data or non-radial sampling patterns?
1. How might Moner’s performance change if the motion were non-rigid or occurred within acquisition frames?
1. What specific benefits does the coarse-to-fine hash encoding offer over traditional encoding techniques for motion correction?
1. What specific attributes of Moner contribute to its improved robustness on OOD data compared to Score-MoCo? Does the implicit neural representation (INR) framework offer an inherent advantage over diffusion-based priors in terms of adaptability to different MRI datasets?

---

> ### Author Response · Authors · 2024-11-21
>
> We sincerely appreciate your time and effort in reviewing our work. Your constructive feedback motivates us to further improve our study. Below, we address your comments in detail.
>
> ---
>
> **W1: Extension to 3D and Non-Radial Sampling Patterns.**
>
> **A1 (_Extention to 3D Radial MRI_)**: We have extended the proposed Moner to the 3D stack-of-radial sampling pattern. In 3D radial MRI, the 2D inverse Fourier transform of a spoke array from a specific view corresponds to its respective 2D projections, enabling the extension of Moner to address the problem of parallel back-projection in 3D space.
>
> To validate this extension, we conducted simulation experiments on a 3D brain MRI volume with dimensions of 240$\times$240$\times$240. As shown in Table R1, our Moner achieves reconstruction quality consistent with the reference images, demonstrating its feasibility and effectiveness for 3D radial MRI. For a more detailed discussion on the extension to 3D radial MRI, please refer to `the revised submission (Sec. A.1)`.
>
> | **AF** | **MR**  | **NuIFFT**          | **Ours**       |
> |--------|---------|---------------------|----------------|
> | 2x     | ±5      | 23.64**/0.499**    | **31.58/0.844** |
> |  2x      | ±10     | 21.77**/0.435**    | **31.64/0.843** |
> | 4x     | ±5      | 23.29**/0.439**    | **30.68/0.810** |
> | 4x            | ±10     | 21.81**/0.391**    | **30.70/0.813** |
>
> _Table R1: Quantitative results of 3D MR image by NuIFFT and our Moner the 3D brain data. Results of t-test statistical tests comparing our Moner to NuIFFT are denoted by $^{**}$ ($p$-value $<$ 0.01), $^{*}$ ($p$-value $<$ 0.05), and $^\blacktriangledown$ (not significant, $p$-value $\ge$ 0.05)._
>
> **A2 (_Non-radial Sampling Patterns, e.g., Cartesian sampling_)**: Our Moner is originally designed for radial MRI, as each spoke in radial sampling passes through the center of k-space. The 1D inverse Fourier transform of a spoke corresponds to a projection integral, allowing the reconstruction of radial MRI to be reformulated as a back-projection problem. In contrast, Cartesian sampling acquires k-space data on a regular grid, where each spoke is parallel to the readout axis (i.e., k_x), and adjacent spokes are separated by fixed intervals along the phase-encoding axis (i.e., k_y). According to Fourier transform theory, a translation in the frequency domain results in a phase shift in the spatial domain. This makes it challenging to apply the current Moner directly to Cartesian sampling.
>
> Technically, one approach to extending Moner to Cartesian sampling is to apply a phase shift to the MR image predicted by the MLP network before integrating it into the projection-based forward model. However, Cartesian sampling lacks the inherent redundancy of low-frequency information present in radial sampling, which could reduce the efficiency and stability of the model's optimization. Therefore, Cartesian sampling with specifically designed trajectories to simultaneously scan high-frequency and low-frequency spokes [1][2] may be required. Moreover, adopting an alternating optimization strategy could help enhance model convergence and performance under Cartesian sampling patterns.
>
> > [1] Levac, Brett, Ajil Jalal, and Jonathan I. Tamir. "Accelerated motion correction for MRI using score-based generative models." 2023 IEEE 20th International Symposium on Biomedical Imaging (ISBI). IEEE, 2023.
>
> > [2] Levac, Brett, et al. "Accelerated motion correction with deep generative diffusion models." Magnetic Resonance in Medicine 92.2 (2024): 853-868.
>
> ---
>
> **W2: Reliance on Motion Assumptions: The quasi-static motion model assumes rigid motion between acquisition frames. Further clarification on its limitations in more complex motion settings could strengthen the study.**
>
> **A**: The quasi-static motion model assumes rigid motion between acquisition frames and is reasonable and practical for most of the clinical applications. However, this model does have some limitations, particularly in its inability to simulate some complex motion situations, including motion within acquisition frames, non-rigid motion, or complicated spin-history effects. Fortunately, there are some potential solutions to these limitations. For motion within acquisition frames, since radial MRI scans take only a few milliseconds to acquire a single spoke, intra-frame motion is usually negligible. For non-rigid motion, we can use deformation field modelling and incorporate it into the optimization process of the Moner network.
>
> We have added this discussion in `the revised submission (Sec. 5)` as below:
>
> _Secondly, the quasi-static motion model assumes rigid motion between acquisition frames, but it falls short in simulating certain complex motion scenarios. These include motion occurring within acquisition frames, non-rigid motion, and intricate spin-history effects. ... Also, for non-rigid motion, Moner can be extended by incorporating deformation field modeling (Reed et al., 2021)._

---

> ### Author Response · Authors · 2024-11-21
>
> **Q1: Could the authors elaborate on potential strategies to adapt Moner for 3D MRI data or non-radial sampling patterns?**
>
> **A**: Please refer to `our response to W1` for additional details on this topic.
>
> ---
>
> **Q2: How might Moner’s performance change if the motion were non-rigid or occurred within acquisition frames?**
>
> **A**: Thank you for the insightful question. The proposed Moner currently employs a quasi-static motion model, which operates under two key assumptions: 1) the motion is rigid, and 2) motion occurs only between spokes, while remaining static within each spoke. Consequently, the current version of Moner cannot directly address non-rigid motion or motion occurring within acquisition frames.
>
> However, the quasi-static model remains both reasonable and practically significant for many clinical applications. Radial MRI scans acquire a single spoke in just a few milliseconds, during which motion is generally negligible, and rigid motion is common in clinical MRI scans of the brain and legs.
>
> To handle non-rigid motion, Moner could be extended by incorporating non-rigid motion models. For instance, in 4D dynamic CT imaging, previous work has introduced deformation fields to address non-rigid motion [3]. Similarly, Moner could be enhanced by integrating deformation field modeling into the optimization process of the MLP network. This approach could allow Moner to estimate and compensate for more complex motion patterns during reconstruction.
>
> > [3] Reed, Albert W., et al. "Dynamic ct reconstruction from limited views with implicit neural representations and parametric motion fields." Proceedings of the IEEE/CVF International Conference on Computer Vision. 2021.
>
> ---
>
> **Q3: What specific benefits does the coarse-to-fine hash encoding offer over traditional encoding techniques for motion correction?**
>
> **A**: Hash encoding assigns learnable features to each position in the imaging space, unlike traditional encoding techniques such as Fourier or position encoding, which rely on pre-defined mathematical transformations. This adaptiveness allows hash encoding to use a compact MLP (typically 2–4 layers) to approximate high-frequency signals effectively, enabling faster convergence and improved reconstruction quality.
>
> However, traditional hash encoding’s strong ability to fit high-frequency signals can sometimes lead to overfitting, negatively affecting motion correction (MoCo) performance. To address this, we introduce a coarse-to-fine strategy that balances detail preservation with MoCo robustness. The strategy starts with a low-frequency approximation in the early stages and progressively refines the reconstruction, allowing better alignment between motion modeling and image reconstruction.
>
> As shown in Table R2, the coarse2fine hash encoding achieves the best performances in both image reconstruction quality and optimization speed. `Figure R2 of the Supplementary Material` shows the quantitative results. These results demonstrate that advanced encoding schemes, particularly coarse-to-fine hash encoding, can significantly enhance both image reconstruction and motion correction in MRI.
>
> | **Metric**               | **Position Encoding** | **Coarse2fine Hash Encoding** |
> |--------------------------|-----------------------|--------------------------------|
> | PSNR                | 30.53±2.34           | **32.37±1.94**                |
> | SSIM                | 0.900±0.026          | **0.935±0.014**               |
> | Average Recon. Time | 10 minutes           | **5 minutes**                 |
>
> _Table R2: Qualitative results of our Moner with two different encoding modules on the fastMRI dataset for AF=2x and MR=±5._

---

> > ### Author Response · Authors · 2024-11-21
> >
> > **Q4: What specific attributes of Moner contribute to its improved robustness on OOD data compared to Score-MoCo? Does the implicit neural representation (INR) framework offer an inherent advantage over diffusion-based priors in terms of adaptability to different MRI datasets?**
> >
> > The core distinction between Moner and Score-MoCo lies in their foundational frameworks. Moner, based on implicit neural representations (INRs), operates unsupervised and data-specific, relying solely on the physics-based forward model and the implicit regularization of neural networks [4]. This approach avoids dependence on pretrained priors, making it inherently robust to OOD scenarios.
> >
> > In contrast, Score-MoCo uses a pretrained diffusion model to provide data distribution priors, which depend heavily on the quality and diversity of the training data. When test data deviates significantly from the learned manifold, Score-MoCo faces generalization challenges [5]. Moner’s independence from data-driven priors eliminates these limitations, enabling consistent performance across datasets with varying characteristics.
> >
> > > [4] Martin, Charles H., and Michael W. Mahoney. "Implicit self-regularization in deep neural networks: Evidence from random matrix theory and implications for learning." Journal of Machine Learning Research 22.165 (2021): 1-73.
> >
> > > [5] Renaud, M., Liu, J., De Bortoli, V., Almansa, A., & Kamilov, U. S. (2023). Plug-and-Play Posterior Sampling under Mismatched Measurement and Prior Models. arXiv preprint arXiv:2310.03546.

---

### Official Review · Reviewer_rXFf · 2024-11-08

**Soundness:** 3
**Presentation:** 4
**Contribution:** 3
**Rating:** 8
**Confidence:** 5

**Summary:**

This paper proposes a novel INR that includes motion modeling, allowing for rigid motion-corrected reconstructions. The motion is modeled using translation and rotation parameters to deform a canonical image space, and then transform the the image using the Fourier slice theorem to compute the data consistency loss with the acquired k-space data.
Evaluation is conducted on simulated motion-corrupted brain data (2 public datasets) and the individual proposed components (motion model, Fourier slice theorem, coarse-to-fine has encding) ablated.

**Strengths:**

The following strengths can be observed within the paper:
- The inclusion of a motion model for MR reconstruction is sound and by leveraging the Fourier slice theorem, the inclusion directly in image-space allows simple motion modeling.
- The work includes extensive evaluation, including several comparison methods as well as reasonable ablation studies.
- The paper is nicely structured.

**Weaknesses:**

The present work poses some major and minor weaknesses:
Major:
- The evaluation is exclusively conducted on simulated motion-corrupted data, which raises the question of its actual applciability in real settings. Particularly considering that motion artefacts not only arise from the physical movement, but also from resulting field inhomogenieties, etc., the simulation procedure might be too simplified and the model is likely to simply adapt to these parameters. If the work is aimed at MR motion-correction methods within a clinical workflow, testing the method on in-vivo data is mandatory.
- In this work, it is claimed multiple times, that the Fourier slice theorem is a crucial part of their work and this works novelty. Yet, the Fourier slice theorem was already introduced a while ago (Catalan 2023 et al,
https://doi.org/10.48550/arXiv.2307.14363) and used in several works after. While it is definitely aiding the method, it should be cited and not included as novelty.
- Within the results, "significant improvement" (l., 365/l.431) is often claimed, the standard deviation (e.g. in table 2) and statistical tests should be reported to support this statement. Similarly for the improvement claim regarding Table 4: Statistical testing would be helpful, since the standard deviation reported opens up questions the actual improvement

Minor:
- For full reproducibility, it would be nice to mention all post-processing steps conducted before quantitative evaluation (nect to registration mentioned in A1, e.g. if normalization was conducted, etc.?)
- There are several typos which can be corrected (line326, 504,...)

**Questions:**

- For the motion simulation, are all spokes corrupted by motion? Or how many spokes are chosen to be affected? Realistic motion patterns in brain would be individual lines, but this is not specified. Can you generally provide more details on the motion simulation, e.g. was only the physical movement considered / what motion patterns were the inspiration for this simulation / were any secondary motion effects considered?
- In line 151 it is mentioned, that the methods overlook the high dynamic range problem, yet it is considered in some of the mentioned works (e.g. Huang 2023)?
- The low-frequency bias mentioned in the introduction (l.182) is highly dependent on the network settings (activation function, etc.), yet this is not mentioned in the implementation details?
- I would suggest to add significance testing to support the result claims within the text.
- In general, while the simulation evaluation is nice for a proof-of-concept, the models performance on real-world data would be very useful to evaluate its actual capabilities and highly recommended for any future work.

---

> ### Author Response · Authors · 2024-11-21
>
> Thank you for the insightful comments. We are encouraged by your recognition of our work. Below, we provide detailed point-by-point responses to address your concerns.
>
> ---
>
> **W1: The evaluation is exclusively conducted on simulated motion-corrupted data, which raises the question of its actual applciability in real settings. Particularly considering that motion artefacts not only arise from the physical movement, but also from resulting field inhomogenieties, etc., the simulation procedure might be too simplified and the model is likely to simply adapt to these parameters. If the work is aimed at MR motion-correction methods within a clinical workflow, testing the method on in-vivo data is mandatory.**
>
> **A1 (Evaluation on Real-world Data):** To evaluate the effectiveness of our Moner on real-world MRI data, we collect two T1w brain 3D scans from a single subject using a 3.0 T United Imaging Healthcare (UIH) uMR 790 scanner. To obtain motion-free data and motion-corrupted data, the subject is instructed to remain still or make abrupt movements three times during acquisition. The data acquisition is approved by the institutional review board. As shown in `Figure 12 in the revised submission`, our Moner reconstruction closely resembles the reference in global structures and successfully recovers tissue details. This study preliminarily validates the effectiveness of our method. We have included this study on real-world data in `the revised submission (Sec. A.1)`.
>
> **A2 (Modeling for Non-movement Factors):** MRI acquisition is a complicated process. We agree with the reviewer that many non-movement factors, such as field inhomogeneities, could affect the quality of reconstructed images. However, this work aims to develop a new method to address the MRI MoCo problem by correcting the subject's movements. We leave these modeling of non-movement factors in our further work.
>
> ---
>
> **W2: In this work, it is claimed multiple times, that the Fourier slice theorem is a crucial part of their work and this works novelty. Yet, the Fourier slice theorem was already introduced a while ago (Catalan 2023 et al, https://doi.org/10.48550/arXiv.2307.14363) and used in several works after. While it is definitely aiding the method, it should be cited and not included as novelty.**
>
> **A:** Thank you for the insightful comment. We acknowledge that the Fourier slice theorem has been introduced in prior works, such as NF-cMRI (Catalan et al., 2023), which reconstructs high-quality cardiac MR images by optimizing a neural field from undersampled radial k-space data. In their work, the Fourier slice theorem is used to transform predicted MR images from the spatial domain into the k-space domain, where the model is optimized.
>
> In contrast, our Moner utilizes the Fourier slice theorem to directly optimize in the projection domain, specifically to address the high-dynamic range problem inherent to k-space data. This approach differs fundamentally in motivation and application. To our knowledge, leveraging the Fourier slice theorem in this manner to overcome the high-dynamic range issue has not been explored in existing literature. As demonstrated in our ablation study, the projection-based optimization significantly enhances MoCo accuracy ($p<0.01$) and moderately improves MRI reconstruction ($p≥0.05$), further validating its effectiveness.
>
> We have cited Catalan et al. (2023) as related work on neural fields for MRI reconstruction in the revised manuscript (line 147) to ensure proper attribution and clarity.

---

> > ### Comment · Reviewer_rXFf · 2024-11-21
> > **Response to W2: Novelty of the Fourier slice theorem**
> >
> > W2: Thank you for clarification of the difference to Catalan 2023 which in my eyes should also be included in the paper to clarify the contributions fairly. To further clarify the difference and not claim that the introduction of the Fourier Slice Theorem is novel, theses crucial differences could be outlined in section 2.1 and also indicated in the introduction.

---

> > > ### Author Response · Authors · 2024-11-21
> > > **Clarifying differences between NF-cMRI and our Moner in using the Fourier-slice theorem.**
> > >
> > > Thanks for your constructive suggestion. We have added relevant statements to clarify the differences between NF-cMRI and our Moner and to fairly highlight contributions (Sec. 2.1), as shown below:
> > >
> > > *A recent study on cardiac MRI acceleration~\citep{catalan2023unsupervised} introduces the Fourier-slice theorem to optimize a neural field directly from raw k-space data, showing promising potential. Similarly, but with a distinct focus, our work uses the Fourier-slice theorem to reformulate radial MRI as a back-projection problem, fundamentally addressing the high-dynamic range issue and stabilizing optimization.*

---

> > > > ### Comment · Reviewer_rXFf · 2024-11-27
> > > >
> > > > I appreciate the author's responses which addressed my main concerns. Therefore, I have raised my score to 8 Accept.

---

> > > > > ### Author Response · Authors · 2024-11-27
> > > > >
> > > > > Thank you for recognizing our work! Your constructive suggestions have greatly enhanced it.

---

> ### Author Response · Authors · 2024-11-21
>
> **W3: Within the results, "significant improvement" (l., 365/l.431) is often claimed, the standard deviation (e.g. in table 2) and statistical tests should be reported to support this statement. Similarly for the improvement claim regarding Table 4: Statistical testing would be helpful, since the standard deviation reported opens up questions the actual improvement**
>
> **A:** Thank you for your constructive suggestions. In our original submission, only mean PSNR and SSIM values were reported due to space limitations. In the revised submission, we now provide comprehensive results, including Mean$\pm$STD and statistical t-tests, in `Tables 2 and 7 of the revised submission`. The key observations are as follows:
>
> - Compared to NuIFFT, TV, and DRN-DCMB, our model shows statistically significant improvements across all cases ($p<0.01$).
>
> - On the fastMRI dataset, the current SOTA method, Score-MoCo (pre-trained on fastMRI), slightly outperforms Moner (by approximately 1 dB in PSNR), but the improvement is not statistically significant ($p≥0.05$).
>
> - On the out-of-domain (OOD) MoDL dataset, our unsupervised Moner achieves steady and statistically significant improvements ($p<0.05$) over Score-MoCo in most cases.
>
> For our ablation studies, the statistical t-tests in `Tables 3, 4, and 5 of the revised submission` reveal the following:
>
> - The motion model is a critical component, yielding significant improvements ($p<0.01$).
>
> - The projection-based optimization significantly enhances MoCo accuracy ($p<0.01$) and moderately improves MRI reconstruction ($p≥0.05$).
>
> - The coarse-to-fine strategy plays an essential role in preventing MoCo degradation from overfitting to high-frequency details.
>
> These statistical analyses and t-tests further validate the superiority of our Moner and underline the importance of its key components. We appreciate your feedback, which has helped us enhance the clarity and rigor of our results.
>
> ---
>
> **W4: For full reproducibility, it would be nice to mention all post-processing steps conducted before quantitative evaluation (nect to registration mentioned in A1, e.g. if normalization was conducted, etc.?)**
>
> **A:** Thank you for the suggestion. In our workflow, post-processing is limited to the registration step mentioned in Sec. A1. While during data pre-processing, the GT images are normalized to the range [0, 1] before the data simulation process. These additional details have been included in `the revised manuscript (Sec. 4.1)` to enhance reproducibility and clarity.
>
> ---
>
> **W5: There are several typos which can be corrected (line326, 504,...)**
>
> **A:** Thank you for your thorough proofreading. We have thoroughly reviewed the manuscript and corrected all identified typos, including those on lines 326 and 504, in the revised submission. We appreciate your attention to detail, which helps improve the overall quality of our work.

---

> > ### Author Response · Authors · 2024-11-21
> >
> > **Q1: For the motion simulation, are all spokes corrupted by motion? Or how many spokes are chosen to be affected? Realistic motion patterns in brain would be individual lines, but this is not specified. Can you generally provide more details on the motion simulation, e.g. was only the physical movement considered / what motion patterns were the inspiration for this simulation / were any secondary motion effects considered?**
> >
> > **A:** Our motion simulation is based on the concept of motion stages [1][2]. Specifically, all spokes are divided into $N$ motion stages, where spokes in the same motion stage share identical motion trajectories. Our Moner model assigns 3 motion parameters per stage and thus totally estimates $N\times$3 unknowns. In our experiment, we use 360 and 180 spokes for acceleration factors (AFs) of 2$\times$ and 4$\times$, respectively. These spokes are divided into $N=18$ motion stages, resulting in the estimation of $18×3=54$ unknowns. We have added this information in `the revised submission (Sec. 4.1)`.
> >
> > Additionally, we evaluate our method across different numbers of motion stages ($N=\{18, 72\}$), corresponding to $18\times 3=54, 72\times 3=216$ unknowns, and simulate different types of motion trajectories (including random, periodic, and abrupt). Note that all hyperparameters for our model remained consistent with those reported in the original submission.
> >
> > As shown in Table R1, our Moner achieves excellent and stable reconstruction performance, indicating minor variations in PSNR ($\pm1$ dB) and SSIM ($\pm$0.2), and statistical tests show no significant differences. Qualitative results are provided in `Figure R3 of the Supplementary Material`, where reconstructed MR images across all conditions appear visually consistent, further confirming the robustness of our method.
> >
> >
> > | Type of Motion   | Motion Stage  | PSNR           | SSIM           |
> > |:----------------:|:----------------:|:----------------:|:----------------:|
> > |       Random       |         N=18         |  32.71±1.03      |   0.907±0.046    |
> > |       Random       |         N=72         |  32.48±1.09 ▼    |   0.904±0.048 ▼  |
> > |      Periodic      |         N=18         |  32.88±1.11 ▼    |   0.910±0.048 ▼  |
> > |      Periodic      |         N=72         |  32.45±0.78 ▼    |   0.906±0.047 ▼  |
> > |       Abrupt       |         N=18         |  32.54±1.40 ▼    |   0.897±0.051 ▼  |
> > |       Abrupt       |         N=72         |  31.75±1.08 ▼    |   0.884±0.058 ▼  |
> >
> >
> > *Table R1: Quantitative results of our Moner on the fastMRI dataset for AF=2x and MR=±5 when simulating different types of motion trajectories and different numbers of motion stages. Results of t-test based statistical tests comoaring Row #1 (Random & N=18) to other settings are presented by \*\*(p<0.01), \*(p<0.05), and ▼ (no significant, p≥0.05).*
> >
> > > [1] Spieker, Veronika, et al. "Deep learning for retrospective motion correction in MRI: a comprehensive review." IEEE Transactions on Medical Imaging (2023).
> >
> > > [2] Levac, Brett, et al. "Accelerated motion correction with deep generative diffusion models." Magnetic Resonance in Medicine 92.2 (2024): 853-868.
> >
> > ---
> >
> > **Q2: In line 151 it is mentioned, that the methods overlook the high dynamic range problem, yet it is considered in some of the mentioned works (e.g. Huang 2023)?**
> >
> > **A:** Thank you for pointing out this vague statement. We have updated the relevant statements in `the revised manuscript (line 150)` as below:
> >
> > _Moreover, these methods either entirely overlook the high dynamic range problem in k-space data or specifically design a relative $\ell_2$ loss to alleviate it._
> >
> > ---
> >
> > **Q3: The low-frequency bias mentioned in the introduction (l.182) is highly dependent on the network settings (activation function, etc.), yet this is not mentioned in the implementation details?**
> >
> > **A:** Thank you for pointing out the issue. We have added detailed descriptions of the network settings in `the revised submission (Sec. 4.1)` as below:
> >
> > _For our Moner, we employ the hash encoding followed by two fully-connected (FC) layers with a width of 128 to implement the INR network. The first FC layer is followed by a ReLU activation, while the second one (i.e., the output layer) has no activation._
> >
> > ---
> >
> > **Q4: I would suggest to add significance testing to support the result claims within the text.**
> >
> > **A:** Thank you for your constructive suggestions. We have added the related results in the revised submission, please refer to `our response to W3` for additional details on this topic.
> >
> > ---
> >
> > **Q5: In general, while the simulation evaluation is nice for a proof-of-concept, the models performance on real-world data would be very useful to evaluate its actual capabilities and highly recommended for any future work.**
> >
> > **A:** Please refer to `our response to W1` for additional details on this topic.

---

> ### Comment · Reviewer_rXFf · 2024-11-21
> **Response to W1: Addition of real-world examples**
>
> Thank you for the valuable addition of a in-vivo motion corrupted case. You mention how subjects were instructed to move. Could you comment on the estimated motion parameters for the motion-corrected reconstruction and if this aligns with your original motion corruption setup?

---

> > ### Author Response · Authors · 2024-11-21
> > **Visualization of estimated motion trajectory on real-world MRI data**
> >
> > During the acquisition, the subject is instructed to make abrupt movements (head-shaking motion) three times.  The visualizations of the motion trajectory estimated by our Moner are shown in `Figure 13 of the revised submission`.  Generally, the estimated trajectory aligns well with our motion pattern setup.  Thank you for your constructive feedback, which helps us enhance our work.

---

### Author Response · Authors · 2024-11-25
**Summary of Revisions and Gentle Reminder**

Dear AC and Reviewers,

We deeply appreciate the engaging discussion, which has significantly improved our work. Below is a summary of the revisions we have made in response to your feedback:

1. Extended Moner to 3D radial MRI and evaluated it on *in-vivo* 3D data (`Sec. A.1`) [rXFf, 63Er, mfmU].
2. Added detailed experimental setups (`Sec. 4.1`) and included statistical tests (`Tables 2 and 7`) [rXFf, mfmU, kTjY].
3. Discussed applications of Moner to non-radial sampling patterns [63Er, kTjY].
4. Clarified ambiguous statements (`Secs. 2.1 and 2.3`) and corrected all typos [rXFf].

With the discussion phase ending on November 26th, we warmly welcome any further comments or suggestions.

Once again, we thank you for your invaluable feedback and support in refining our work.

Best regards,

The authors

---

### Meta-Review · Area_Chair_yymR · 2024-12-06

**Metareview:**

The paper proposes a method for reconstructing magnetic resonance images and estimating motion parameters from undersampled motion-corrupted measurements. The approach is based on implicit neural networks.

The paper got four detailed reviews. All significant weaknesses (like details on the simulation of the motion artifacts, evaluation on real subjects etc) were addressed during the rebuttal. After the rebuttal, all four reviewers recommend to accept the paper, and I also recommend accepting the paper.

**Additional Comments On Reviewer Discussion:**

The reviewers had concerns about lacking evaluations on real data, phrasing of the contributions, and details on the simulation of the simulated motion. Those were addressed in the rebuttal by adding evaluations on real data as well as revising the paper.

---

### Decision · Program_Chairs · 2025-01-22

Accept (Spotlight)